# Important role of endogenous microbial symbionts of fish gills in the challenging but highly biodiverse Amazonian blackwaters

Sylvain François-Étienne [1,2] ✉, Leroux Nicolas[1], Normandeau Eric [1], Custodio Jaqueline [3], Mercier Pierre-Luc[1], Bouslama Sidki[1], Holland Aleicia[4], Barroso Danilo[3,5], Val Adalberto Luis [3] & Derome Nicolas[1]

Amazonian blackwaters are extremely biodiverse systems containing some of Earth's most naturally acidic, dissolved organic carbon -rich and ion-poor waters. Physiological adaptations of fish facing these ionoregulatory challenges are unresolved but could involve microbially-mediated processes. Here, we characterize the physiological response of 964 fish-microbe systems from four blackwater Teleost species along a natural hydrochemical gradient, using dual RNA-Seq and 16 S rRNA of gill samples. We find that host transcriptional responses to blackwaters are species-specific, but occasionally include the overexpression of Toll-receptors and integrins associated to interkingdom communication. Blackwater gill microbiomes are characterized by a transcriptionally-active betaproteobacterial cluster potentially interfering with epithelial permeability. We explore further blackwater fish-microbe interactions by analyzing transcriptomes of axenic zebrafish larvae exposed to sterile, non-sterile and inverted (non-native bacterioplankton) blackwater. We find that axenic zebrafish survive poorly when exposed to sterile/inverted blackwater. Overall, our results suggest a critical role for endogenous symbionts in blackwater fish physiology.

Blackwater systems are found globally from peatlands in European peat bogs, to wallum streams in Australia, swamps along the Atlantic coast of North America, and South American rainforests[1–3]. In the Amazon basin, blackwater systems contain some of the most naturally acidic, dissolved organic carbon (DOC)-rich and ion-poor waters anywhere in the world[4]. Their pH can be as low as ~3.0, due to their poor buffering capacity as well as the high input of decaying tropical plant material from the dense surrounding forest[5], resulting in a high humic DOC concentration, typically 5–15 mg/L[6]. These DOC-rich waters harbor relatively high concentrations of metals (Fe, Al, Zn) reflecting the preferential binding of metals to humic substances[7] and the changes to metal speciation associated with low pH. However, the binding properties of the silicate sand found at the bottom of these habitats strips the water from its major cations ($Na^+$, $Cl^-$, $Mg^{2+}$, and $Ca^{2+}$), leaving it extremely ion-poor[7]. Overall, the chemistry of these blackwaters appears so extreme that they would most likely be toxic to most freshwater fish, yet the ichthyofauna inhabiting these habitats is one of the most biodiverse on Earth, with an estimated richness of 1700 species[8], surpassing the estimated 1620 fish species on the Great Barrier Reef[9]. Thus, this biodiversity inevitably possesses a range of physiological adaptations to this otherwise very challenging environment.

[1]Institut de Biologie Intégrative et des Systèmes, Université Laval, 1030 avenue de la Médecine, Québec (QC) G1V 0A6, Canada. [2]Fisheries and Oceans, Gulf Fisheries Center, 343 University Ave, Moncton, NB E1C 5K4, Canada. [3]Instituto Nacional de Pesquisas da Amazônia (INPA), Laboratório de Ecofisiologia e Evolução Molecular, Manaus, AM 69067-375, Brazil. [4]La Trobe University, School of Agriculture, Biomedicine and Environment, Department of Environment and Genetics, Centre for Freshwater Ecosystems, Albury/Wodonga Campus, Vic, Australia. [5]Deceased: Barroso Danilo. ✉e-mail: francois-etienne.sylvain.1@ulaval.ca

The general physiological response of fish to acidic and ion-free waters is the inhibition of active $Na^+$ and $Cl^-$ uptake and increase of passive ion loss resulting in reduced plasma $Na^+$ and $Cl^-$ levels, lowered blood volume, haemoconcentration, and elevated plasma protein concentrations[10], combined to a leaching of $Ca^{2+}$ ions disturbing the epithelial integrity of tight junctions[11], which ultimately leads to mortality due to circulatory failure[12]. Amazonian blackwater fish species show an entirely different response: They possess an exceptionally high acid tolerance, surviving at or below pH 4.0 by avoiding imbalances in net $Na^+$ and $Cl^-$ flux rates[6]. Aside from a general resistance to the stimulation of gill diffusive ion loss caused by low pH, one common feature among blackwater fish may be the maintenance of generally lower plasma $Na^+$ levels, a strategy to minimize energy expenditure in low $Na^+$ environments[13]. However, the degree of tolerance to extremely dilute low pH waters is species-specific[14]: There appears to be two basic ionoregulatory strategies of fish native to ion-poor, acidic blackwaters. For instance, a high affinity ( = low $K_m$) ion transport system that supports high rates of $Na^+$ uptake is characteristic of the Characiformes, and a very tight regulation of $Na^+$ efflux at low pH is typically observed in the Cichliformes[15]. The processes underlying these ionoregulatory strategies are still not understood, however, they appear to be unique and do not fit current models from non-native fish[6].

Several studies suggest that DOC is involved in some way in these strategies. Although the dissociation of protons from allochthonous humic DOC is the major cause of acidity in natural blackwaters, evidence from different fish clades shows that these substances could mitigate the ionoregulatory disturbances of the low pH which they help to create[16,17]. Humic DOC is an important regulator of biotic and abiotic processes in freshwater ecosystems[18,19]. Campbell et al.[20] demonstrated, by transmission electron microscopy, that DOC molecules can bind directly to the surface of gill cells. In fish exposed to acidic waters, DOC mitigates the inhibition of $Na^+$ uptake and ammonia excretion[19], lowers the transepithelial potential (TEP)[21], protects against the elevation of diffusive $Na^+$ and $Cl^-$ loss[16,17,19] and reduces the gill binding and toxicity of metals[22]. Humic DOC has also been shown to significantly decrease respiratory stress caused by exposure to low pH and increase the survival of fish at pH 4[23]. However, some DOCs can also have toxic effects on fish at low pH[24] or have a hormesis effect whereby protective only at certain concentrations or pHs[23]. Recent studies suggest that the protective roles of DOC depend on its chemical characteristics or integrity: Humic DOC exacerbated rather than protected against ionoregulatory disturbance both in zebrafish (*Danio rerio*) and Amazonian tambaqui (*Colossoma macropomum*)[25,26]. Overall, the protective role of humic DOC is promising, but current results are contradictory and the processes involved are still unresolved.

One aspect neglected in physiological studies focused on blackwater fish ionoregulatory strategies so far is the role of the gill microbiome. Mucosal microbiota are known to tightly regulate their host's epithelial barrier function[27]. Studies on mice, humans, and cell cultures demonstrated that exposition of epithelial cells to probiotic or commensal bacteria metabolites benefits barrier integrity by regulating epithelial permeability[28–30]. Bacteria target various intracellular pathways, change the abundance, expression, and distribution of tight-junction proteins at the cell boundaries, and thereby regulate membrane permeability[31]. They also utilize epithelial-cell signaling proteins involved in tight-junction regulation, including Rho family GTPases, protein kinase C (PKC), and mitogen-activated protein kinase (MAPK) to enhance barrier integrity[32]. Studies focused on fish gill diseases have shown that bacteria can modulate gill epithelial integrity[33]. The epithelial permeability of fish gills also depends on tight-junction complexes, which play critical roles in the ionoregulatory response of fish exposed to acidic and ion-poor water[34]. A recent study on the medaka fish model[35], integrating transcriptomics and metagenomics to characterize the response of fish holobionts to hypotonic stress, suggested that host-bacterium interactions could facilitate the freshwater acclimation through the glycosaminoglycan and chitin pathways, but the functions involved remain unknown, as the study inferred functional profiles with PICRUSt2 and not through a dual RNA-Seq approach. One question remains: Are gill bacterial symbionts implicated in the physiological response of fish to blackwater habitats?

In this work, we report on the responses of Amazonian fish and their associated gill microbial communities to blackwater environments. We first examined fish and microbial transcriptional activity from over 900-gill samples from four native Amazonian blackwater teleosts collected across a hydrochemical gradient of 15 sampling sites comprising natural blackwater and non-blackwater (white and clearwater) environments which show circumneutral pH, lower DOC concentrations, and higher conductivity levels[4]. We characterized the physiological response of each host species along the gradient and assessed its correlation with gill microbiome transcriptional activity. Then, we explored further the dynamics of blackwater fish-microbe interactions by analyzing the transcriptomes of axenic zebrafish larvae exposed to sterile, non-sterile, and inoculated (non-native bacterioplankton) blackwater in the laboratory. Overall, we provide a framework for understanding ionoregulatory processes of blackwater fish at the holobiont level.

## Results and discussion

### Amazonian fish host physiological response to blackwater

The physiological responses of Amazonian fish to blackwater environments are still currently unresolved. In this study, we addressed this topic by sequencing the gill transcriptomes of four widely distributed Amazonian teleost species (*Mesonauta festivus*, *Cichla* spp., *Triportheus albus*, and *Serrasalmus rhombeus*) collected along a hydrochemical gradient of 15 sampling sites (Supplementary Table 1) comprising five blackwater sites and 10 sites of white- or clearwater (sequencing output summarized in Supplementary Table 2). Overall, differential gene expression (DGE) analyses on transcripts from genes involved in ionoregulation processes, between black- and non-blackwater specimens, showed a species-specific physiological response to blackwater (Fig. 1).

The gill transcriptomes of the two Cichliformes sampled in this study (*M. festivus* and *Cichla* spp.) revealed different strategies to cope with acidic blackwater. The response of *M. festivus* to blackwater included the overexpression of genes involved in the transport of $Ca^{2+}$ (the calcium-transporting ATPase type 2 C member 1 (ATP2C1) and the transient receptor potential cation channel subfamily V member 5 (TRPV5)), $Zn^{2+}$ (the solute carrier family 30 member 8 (SLC30A8) and the transient receptor potential cation channel subfamily M member 2 (TRPM2)), and the $H^+$-dependant transport of $Na^+$ (the sodium/hydrogen exchanger 2 (NHX2), the sodium–hydrogen antiporter 3 (SLC9A3) and the sodium–hydrogen exchange regulatory cofactor 2 (SLC9A3R2)). Whether $H^+$ is required as a counter-ion for $Na^+$ uptake remains unclear in the current literature on native blackwater fish[36]. However, this appears to be the case in *M. festivus* as three $Na^+/H^+$ exchangers were overexpressed in blackwater. Relying on $Na^+/H^+$ exchangers could represent an efficient strategy to sustain gill ion uptake in $Na^+$-poor and $H^+$-rich blackwaters. The response of *M. festivus* also included the overexpression of the gap junction beta-2 protein (GJB2), also known as connexin 26 (Cx26). GAP junctions are aggregates of intercellular channels that permit direct cell–cell transfer of ions and small molecules[37], thereby buffering spatial gradients of ions and with the potential to regulate localized ion efflux in gills exposed to ion-poor blackwater. Finally, the claudin 8 (CLDN8), located at paracellular junctions, was also overexpressed in blackwater *M. festivus*. Claudins regulate the permeability of tight junctions, a complex of transmembrane and peripheral proteins tethered to the cytoskeleton (reviewed in 34), and thus potentially play a central role in limiting ion efflux from the host.

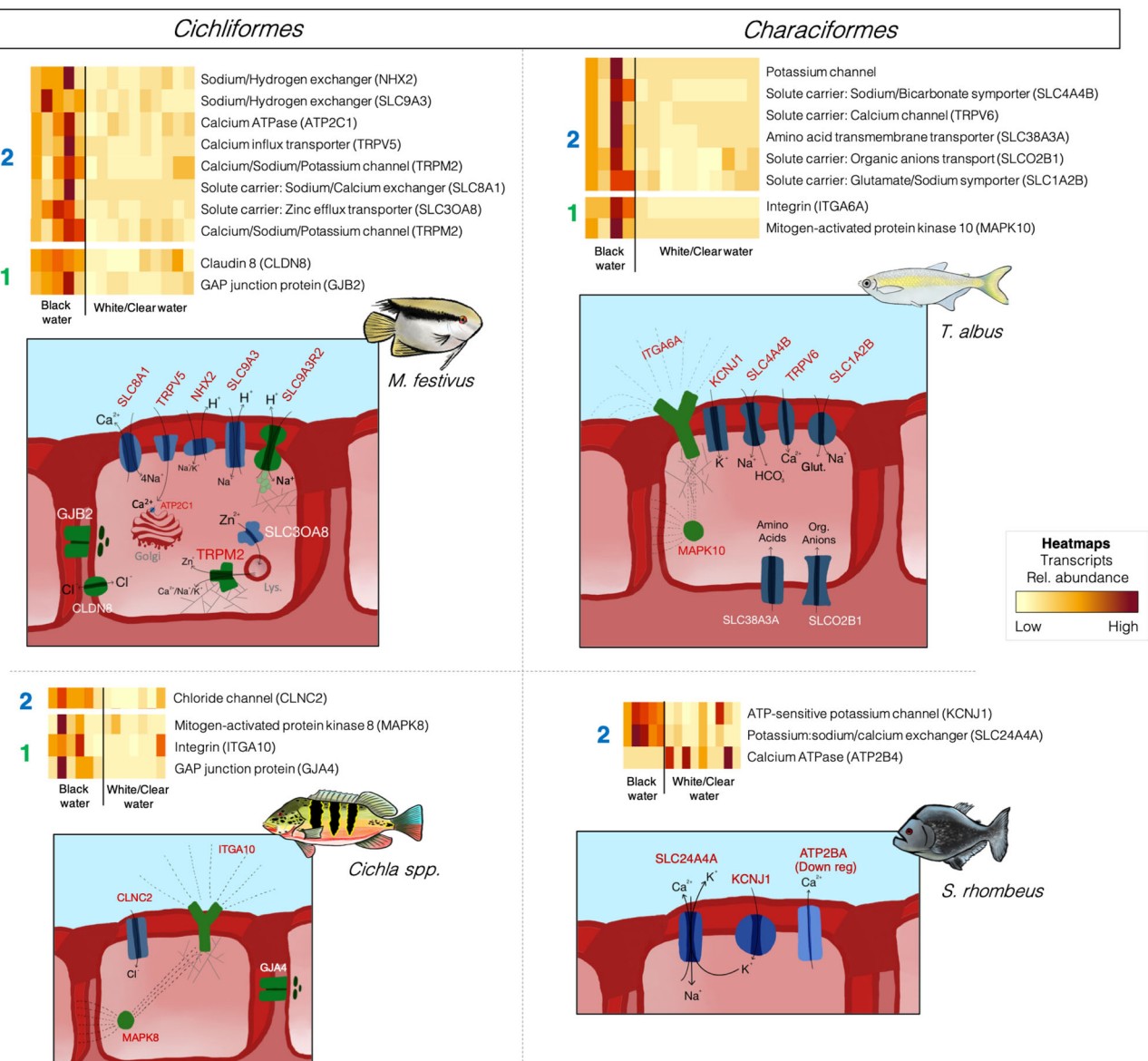

**Fig. 1 | Differential gene expression (DGE) analyses on transcripts from genes involved in ionoregulation processes, between black- and non-blackwater (clear- and whitewater) specimens (dual-RNASeq dataset).** DGE results are shown in heatmaps. For ease of viewing, only transcripts overexpressed in blackwater specimens are shown (except for *S. rhombeus* where underexpressed transcripts were also shown due to the low number of differentially expressed genes identified for this species). In the heatmaps, each row represents a gene while each column is a sample (pool of ~20 fish per species per sampling site). Numbers on the left of each heatmap indicate the ionoregulatory strategy associated with the subset of genes, either (#1) the regulation of ion efflux or (#2) the regulation of active ion uptake. Drawings below each heatmap represent the subset of overexpressed proteins in blackwater and proteins are colored according to the ionoregulatory strategy in which they are involved (#1 in green and #2 in blue). Cellular localization of the proteins and the direction of ion transport may differ from drawing. Dashed lines between proteins indicate potential interactions in signaling pathways. Web-like structures annexed to ITGA10, ITGA6A, and TRPM2 indicate potential interactions with the cytoskeleton. "Golgi" stands for the Golgi apparatus and "Lys." is for lysosome. In the heatmaps, yellow/beige colors are associated with a low relative abundance of the transcripts from the gene in a sample, while dark red colors are associated with high relative abundance. Source data are provided as a Source Data file.

In contrast to *M. festivus*, the response to blackwater of *Cichla* spp. was characterized by an increased regulation of membrane permeability that was not coupled to an overexpression of several ion transport systems (only the chloride channel CLNC2 showed significant DGE) as in *M. festivus*. In *Cichla* spp., the GAP junction protein alpha 4 (GJA4) and mitogen-activated protein kinase 8 (MAPK8) were overexpressed. Mitogen-activated protein kinases are transducers of osmosensory signals in fish gill cells[38] by playing central roles in signaling pathways in osmotic stress responses, including the regulation of intracellular levels of organic osmolytes and inorganic ions[39]. MAPK cascades are also involved in cytoskeletal rearrangements[40] which, in hypotonic environments like blackwater, can translate into swelling of the cytoplasmic compartment accompanied by a reduction in the length of the microvillar diffusion barrier, allowing osmolyte efflux and regulatory volume decrease[41].

Furthermore, the integrin subunit alpha 10 (ITGA10) was overexpressed in *Cichla* spp. from blackwater. Integrins are located at the gill cell surface and are known to interfere with membrane permeability by interacting with Na$^+$/H$^+$ and Na$^+$/Ca$^{2+}$ exchangers[42], or with MAPK signaling pathways[43]. Previous studies have detected an

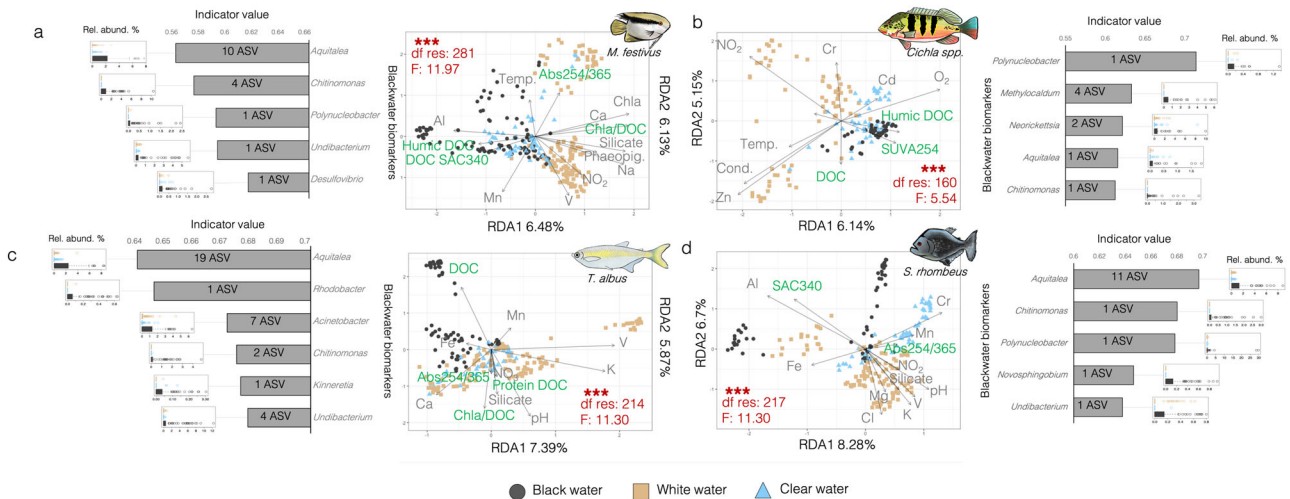

**Fig. 2 | Bray–Curtis distance-based redundancy analyses (dbRDA) on the gill bacterial community samples of each host species (16 S rRNA dataset).** The analysis was conducted on samples from flag cichlids (**a**), peacock bass (**b**), freshwater sardines (**c**), and black piranhas (**d**). Each data point in the dbRDAs represents a microbiome sample colored according to its water type of origin (black for blackwater, beige for whitewater, and blue for clearwater samples). The environmental parameters represented in the dbRDAs were selected by *ordistep* and had a VIF < 10. Environmental parameters colored in green are those associated with dissolved organic carbon quantity or the relative proportion of its different fluorescent fractions (humic-, fulvic- or protein-like). PERMANOVA results (two-sided test) exploring the differences in community composition between water types are colored in red on the dbRDAs. "***" stands for PERMANOVA *p* value < 0.001. The relative abundance (%) of bacterial blackwater biomarkers identified by a multi-level pattern analysis is shown in boxplots (same color key as for the dbRDA), and the average indicator value of the genera represented by those biomarker ASVs is shown in barplots. For this analysis, biologically independent samples of black piranha (N = 231), flag cichlid (N = 296), peacock bass (N = 172), and freshwater sardine (N = 228) microbiotas were examined. The center of the boxplots corresponds to the median. The lower and upper bounds of boxes represent the first and third quartiles (the 25th and 75th percentiles). The lower whiskers extend from the lower bounds of boxes to the lowest values within 1.5 * the inter-quartile range (distance between the first and third quartiles). The upper whiskers extend from the upper bounds of boxes to the highest values that is within 1.5 * the inter-quartile range. Data beyond the end of the whiskers are outliers and plotted as points. Source data are provided as a Source Data file.

overexpression of integrins in fish exposed to hypotonic environments[35,44]. Interestingly, integrins are also involved in interkingdom interactions with environmental bacteria as they can act as pattern recognition receptors for environmental bacteria[45] and can facilitate extracellular adhesion of microorganisms to host cells[46]. Microorganisms can affect the membrane permeability of the host cell[47] and thus the recruitment of permeability-regulating microorganisms via integrin-mediated interactions could potentially benefit fish hosts in the hypotonic blackwater.

The two Characiformes sampled in this study (*S. rhombeus* and *T. albus*) also showed contrasting responses to blackwater. The response of *S. rhombeus* was characterized by the overexpression of an ion transport system (the potassium inwardly-rectifying channel subfamily J member 1 (KCNJ1) and the solute carrier family 24 member 4a (SLC24A4A)) securing the $Ca^{2+}$-dependant exchange of $K^+$ and $Na^+$ across the cellular membranes, combined to a downregulation of the $Ca^{2+}$ plasma membrane export ATPase 4 (ATP2B4).

In *T. albus*, a more complex response was observed, including the overexpression of genes involved in the transport of $Na^+$, $K^+$, $Ca^{2+}$ (KCNJ1, the sodium bicarbonate cotransporter family 4 member 4b (SLC4A4B), and the transient receptor potential cation channel subfamily V member 6 (TRPV6)) as well as transport systems of amino acids such as glutamate (the solute carrier family 38 member 3a (SLC38A3A), the solute carrier organic anion transporter family member 2B1 (SLCO2B1) and the amino-acid transmembrane transporter SLC1A2B). Glutamate can interfere with eukaryotic membrane permeability, can be synthetized by both the host and its microbial symbionts, and can play signaling roles in host-microbe interactions[48]. In addition, as observed in *Cichla* spp., *T. albus* showed the simultaneous overexpression of the integrin subunit alpha 6 (ITGA6A) and mitogen-activated protein kinase 10 (MAPK10) which interact via MAPK signaling pathways. A previous investigation focused on the transcriptomes of *T. albus* showed enrichment of immunity genes in

blackwater specimens, which concord with the overexpression of integrins observed in our study[49].

Overall, the results from these four species (Fig. 1) do not support the evolutionary convergence of physiological adaptation to blackwater, since each species responded differently to blackwater regardless of its order (with the exception of the overexpression of GAP junction proteins in both Cichliformes). Furthermore, the results challenge the current literature[15] stating that Cichliformes rely on tight regulation of ion efflux while Characiformes rely on high-affinity ion uptake systems as both ionoregulatory strategies were observed on three species.

### Gill bacterial communities' response to blackwater

To characterize the response of Amazonian fish gill microbial communities to blackwater, we first used a metabarcoding approach based on the sequencing of the V3-V4 hypervariable 16 S rRNA region in all gill RNA extractions (mean of 19,734 reads per sample, N = 927 samples). The same approach was also carried on bacterioplankton samples, which were characterized in ref. 50 (Supplementary Fig. 1). A general characterization of the active gill bacterial communities of the four Amazonian teleosts showed that they were rich in Betaproteobacteria (mean of 38.85%), Alphaproteobacteria (mean of 9.60%), Flavobacteria (mean of 8.68%), Gammaproteobacteria (mean of 8.30%) and Clostridia (mean of 4.14%) (Supplementary Fig. 2). These results are in accordance with a previous investigation describing the gill microbiota from 53 fish species which identified Proteobacteria and Firmicutes as one of the most abundant phyla on fish gills[51]. Bray–Curtis distance-based redundancy analyses (dbRDA) showed that gill microbiota samples of each host species clustered according to their water type of origin (species-specific analyses on Fig. 2, all species combined in Supplementary Fig. 3). Permutational analyses of variance (PERMANOVA) confirmed that the gill bacterial communities were significantly different between host

species ($p$ val <0.001). A significant effect of host-specific factors on gill bacterial assemblages has been previously detected on large-scale investigations on saltwater fish by Pratte et al.[51] and most recently in a study by Minich et al.[52] which encompassed 101 species. This last study[52] also detected a significant correlation between alpha diversity measures and fish weight or trophic level, however, we did not detect this correlation in our study (Supplementary Fig. 4). Due to species-specific microbiome composition, the response to water type from different host species was analyzed separately (Fig. 2). The two first axes of the dbRDA species-specific ordinations explained between 11.29% and 14.98% of the variance depending on the host species. The fitting of environmental variables on the dbRDA suggested that the quantity of dissolved organic carbon (DOC) and the relative abundance of its different fluorescent fractions (humic, fluvic, or protein-like) were important environmental parameters shaping the gill bacterial communities (see parameters colored in green on Fig. 2). An important role of DOC-related parameters in structuring microbial communities was also found in the free-living bacterioplankton samples collected at these sites (detailed analyses in ref. [50]). In this case, changes in gill bacterial communities according to water types could be associated with an interaction between host-associated and free-living bacterioplanktonic communities specific to each water type (as documented in ref. [50]) or they could also occur in direct response to physicochemical parameters variations. These two environment-associated factors might also explain the significant differences in the gill microbiome from fish of different sampling sites ($p$ val <0.001, Supplementary Fig. 5), supporting previous studies that revealed the sensitivity of gill microbial assemblages to environmental factors (e.g., levels of sediments, salinity levels)[35,53].

PERMANOVAs showed that gill bacterial communities were significantly different between water types ($p$ val <0.001) for all host species considered (see PERMANOVA results in red in Fig. 2). Significant differences in Amazonian fish external (skin mucus) microbiome communities according to water type were previously shown in two studies also focused on *M. festivus* and *S. rhombeus*[54,55].

A multi-level pattern analysis revealed which ASVs were the most strongly associated with blackwater gill microbiomes (i.e., "blackwater biomarkers"). The biomarkers represented several genera of Betaproteobacteria (*Aquitalea, Vogesella, Chromobacterium, Chitinomonas, Polynucleobacter, Undibacterium, Kinneretia*) and a few Alphaproteobacteria (*Novosphingobium, Rhodobacter*), Desulfovibrionia (*Desulfovibrio*), Gammaproteobacteria (*Acinetobacter*) and Flavobacteriia (*Chryseobacterium*). Of note, *Acinetobacter* and *Novosphingobium* were identified as potential contaminants of microbiome studies in the literature[56,57], thus the results pertaining to these taxa should be interpreted with caution. Of all biomarkers identified, only *Polynucleobacter* rRNA transcripts were significantly more abundant in blackwater bacterioplankton samples (Supplementary Fig. 6; also detailed in[50]). All gill blackwater biomarkers showed very low rRNA transcript relative abundances in bacterioplankton (<0.27% in bacterioplankton but up to 29.82% in certain gill samples) suggesting a high niche preference for gills. In this regard, several strains of Betaproteobacteria are known to be endosymbionts of fish gills[58]. A study on Atlantic salmon showed a major enrichment of Betaproteobacteria in microbiomes characterized by whole gills instead of gill swabs[59] suggesting the existence of Betaproteobacterial symbionts within cryptic tissue locations, such as beneath the surface epithelium. Our study showed that rRNA transcripts from the Betaproteobacterial genus *Aquitalea* were particularly differentially abundant in blackwater: On three out of the four Amazonian host species considered, *Aquitalea* ASVs had the highest average indicator value (Fig. 2). *Aquitalea* is known to be involved in host-microbe interactions via its production of indole[60]. Bacterial indole has been shown to modulate a variety of host metabolic and immunological processes[61]. An interesting possibility is that bacterial indole could modulate host ion channels[62] and

increase the host epithelial cell tight-junction resistance[63]. Additional experiments are needed to test to what extent bacterial indole could potentially be involved in the response of fish-microbe systems to blackwater environments.

## Correlations between host and bacterial responses

We investigated whether correlations existed between host and microbial transcriptional activity in blackwater. To do so, we analyzed bacterial RNA-Seq transcriptomes from the blackwater biomarkers identified by the multi-level pattern analysis (Fig. 2) and assessed the Spearman correlation between the abundance of their transcripts and those from differentially expressed host genes shown in Fig. 1. We found six significant correlations with $R^2$ varying between 0.56 to 0.84 (values displayed in Fig. 3). In *S. rhombeus* (Fig. 3a), the bacterial expression of ArgT, a lysine/arginine/ornithine ABC transporter, by *Aquitalea* was correlated to the host expression of two ATP-sensitive K$^+$ transporters (SLC24A4A and XM_017709072.1). Analogously, the host expression of an ATP-sensitive K$^+$ transporter (XM_017709073.2) by *T. albus* was correlated to the bacterial expression of glutP/glutR (a glutamate transport system) by *Undibacterium* and *Acinetobacter*. Glutamate and ornithine/arginine/lysine are closely related: In bacteria, the arginase pathway hydrolyzes arginine to ornithine, which is then hydrolyzed by ornithine aminotransferase (RocD) and Δ-pyrroline-5-carboxylate dehydrogenase (RocA), leading to the production of glutamate[64]. The bacterial production of ornithine promotes the homeostasis of the host epithelial mucus layers[65] by stimulating the tryptophan metabolism to produce AhR ligands which have been shown to correlate with the expression of ATP-sensitive K$^+$ transporters[66].

In *S. rhombeus*, the host expression of the ATP-sensitive K$^+$ transporter (XM_017709072.1) was also correlated to the bacterial expression of porB, a major outer membrane protein (porin), by *Aquitalea* and *Vogesella* (Fig. 3). Interestingly, in another host species considered in this study (*T. albus*), the expression of porB was also correlated to the host expression of the integrin ITGA6A. PorB is expressed by Neisseriales and has the unusual feature of translocating from the outer membrane of the bacteria into artificial and host cell membranes[47]. The insertion process leads to the formation of a functional channel regulated by the eukaryotic host cell[67]. Similar to mitochondrial porins, PorB interacts with purine nucleoside triphosphates which downregulates the pore size, causing a shift in voltage dependence and ion selectivity and also triggers Ca$^{2+}$ influx into the cells[68]. In our study, the fact that the bacterial expression of porB correlated with the host expression of an integrin, a protein involved in cross-kingdom communication, suggests the existence of a potential host-microbe interaction between *Aquitalea* and *T. albus*.

Finally, we observed a correlation between the host expression of a Na$^+$/H$^+$ exchanger by *M. festivus*, and the bacterial expression of atpF, a bacterial H$^+$-ATPase expressed by several taxa including *Undibacterium, Chryseobacterium* (identified as a potential microbiome contaminant in[56,57]), in addition to unidentified Neisseriales and Burkholderiales (Fig. 3c). An exchange of protons could potentially occur between microbial and *M. festivus* cells, however, additional data would be needed to support this hypothesis.

## Responses to blackwater in sterile environments

To better understand how bacteria might interact with their host fish response to blackwater, we conducted a laboratory experiment with axenic zebrafish larvae exposed to black- and whitewater in four treatments[1]: non-sterile water (i.e., with native bacterioplankton)[2], inverted non-sterile water (i.e., with bacterioplankton and sediments from the opposite water type)[3], sterile water (i.e., no bacterioplankton)[4], inverted sterile water (i.e., same as group #2 but sterile). Survival results from the different treatments (Table 1) suggest that the best survival occurred in all treatments of whitewater, followed

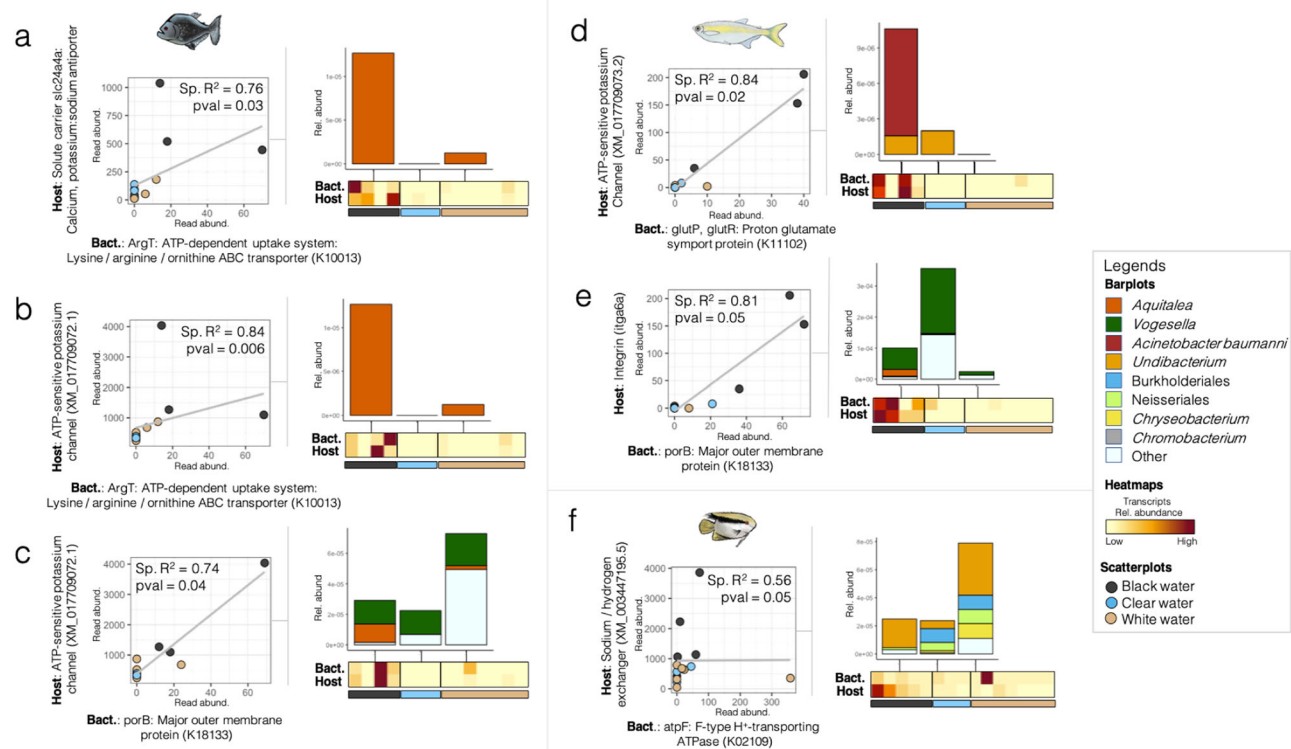

**Fig. 3 | Correlations between the abundance of the transcripts from bacterial RNA-Seq transcriptomes from the blackwater biomarkers (identified in Fig. 2) and those from differentially expressed host genes shown in Fig. 1 (dual RNA-Seq dataset).** The abundance of host and bacterial transcripts is displayed on scatterplots and on the annexed heatmaps, for black piranhas (a–c), freshwater sardines (d, e), and flag cichlids (f). Samples in the scatterplots are colored according to their water type (black for blackwater, beige for whitewater, and blue for clearwater samples). The different taxa producing the bacterial transcripts detected are shown in stacked barplots along the heatmaps. In the scatterplots,

"Sp. $R^2$" stands for Spearman correlation $R^2$. The term "$p$ val" stands for the "$p$ values of the two-sided Spearman correlation tests", which have the following values in **a** $p$ val = 0.03, **b** $p$ val = 0.006, **c** $p$ val = 0.04, **d** $p$ val = 0.02, **e** $p$ val = 0.05 and **f** $p$ val = 0.05. $Y$ axes are shown in log scale to increase visibility. In the stacked barplots, the bacterial taxa are annotated to the best taxonomic resolution possible. In the heatmaps, yellow/beige colors are associated with a low relative abundance of the transcripts from the gene in a sample, while dark red colors are associated with high relative abundance of the transcripts. Source data are provided as a Source Data file.

**Table 1 | Characteristics of the different treatments and mortality rates for the axenic zebrafish experiment***

| Treatment | Water type | Sterility | Bacterioplankton type | Mortality at 18 h (%) |
|---|---|---|---|---|
| BWN | Blackwater | Not sterile | Autochthonous | 4 ± 1 |
| BWI | Blackwater | Not sterile | Allochthonous | 14 ± 3* |
| BWS | Blackwater | Sterile | - | 19 ± 3* |
| BWIS | Blackwater | Sterile | - | 19 ± 2* |
| WWN | Whitewater | Not sterile | Autochthonous | 2 ± 2 |
| WWI | Whitewater | Not sterile | Allochthonous | 1 ± 1 |
| WWS | Whitewater | Sterile | - | 1 ± 1 |
| WWIS | Whitewater | Sterile | - | 1 ± 1 |

*BWN stands for "non-sterile blackwater with native bacterioplankton"; BWI for "inverted non-sterile blackwater" (with whitewater bacterioplankton); BWS for "sterile blackwater"; BWIS for "inverted sterile blackwater" (with sterilized whitewater bacterioplankton); WWN stands for "non-sterile whitewater with native bacterioplankton"; WWI for "inverted non-sterile whitewater" (with blackwater bacterioplankton); WWS for "sterile whitewater"; WWIS for "inverted sterile whitewater" (with sterilized blackwater bacterioplankton). Autochthonous bacterioplankton refers to bacterioplankton from the same water type (e.g., whitewater bacterioplankton in whitewater), while allochthonous bacterioplankton refers to bacterioplankton from the opposite water type (e.g., whitewater bacterioplankton in blackwater). Mortality rates were calculated at the time of sampling after 18 h of exposition to the different treatments. Mortality rates are mean values and their standard deviation from the two replicates of the two times the experiment was conducted (four observations). Mortality rates followed by the symbol (*) indicate treatments where mortality rates reached >90% after 24 h of exposure during preliminary tests prior to the experimentations (data not shown).

by non-sterile blackwater, inverted non-sterile blackwater, with the lowest survival rate occurring in inverted sterile blackwater and sterile blackwater. These results reveal that[1] better larvae survival rates are observed in blackwater in the presence of bacterioplankton than in its absence, and that[2] better larvae survival rates are observed in blackwater inoculated with autochthonous blackwater bacterioplankton rather than with allochthonous (whitewater) bacterioplankton.

After exposure to the different treatments, we sequenced the larvae transcriptomes (sequencing results summarized in Supplementary Table 3) and performed DGE analyses between black- and whitewater treatments to highlight genes potentially involved in ionoregulatory processes (Fig. 4). We found a set of 31 differentially expressed genes between non-sterile black- and whitewater treatments (Fig. 4a, b). Of those, nine genes were also differentially

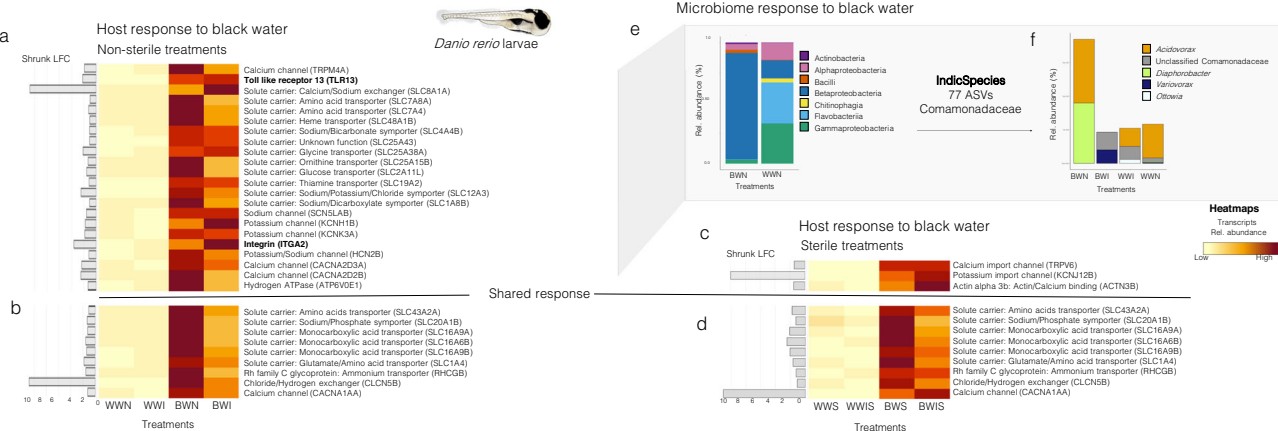

**Fig. 4 | Response of zebrafish host-microbe systems to black- and whitewater sterile and non-sterile treatments. a, b** Heatmap showing the transcript abundances of overexpressed genes in zebrafish larvae exposed to white- and blackwater non-sterile treatments. **c, d** Same as **a, b** but in sterile treatments. **b, d** The shared response is constituted of genes that are overexpressed in both non-sterile and sterile treatments. **e** Stacked barplot showing the relative abundance of the bacterial classes detected on the heads of zebrafish larvae in non-sterile black- and whitewater treatments (16 S rRNA dataset). **f** Stacked barplot showing the relative transcript abundance of the different Comamonadaceae taxa detected in our dataset (dual RNA-Seq dataset). In the stacked barplots, the bacterial taxa are annotated to the best taxonomic resolution possible. BWN stands for "non-sterile blackwater with native bacterioplankton"; BWI for "inverted non-sterile blackwater" (with whitewater bacterioplankton); BWS for "sterile blackwater"; BWIS for "inverted sterile blackwater" (with sterilized whitewater bacterioplankton); WWN stands for "non-sterile whitewater with native bacterioplankton"; WWI for "inverted non-sterile whitewater" (with blackwater bacterioplankton); WWS for "sterile whitewater"; WWIS for "inverted sterile whitewater" (with sterilized blackwater bacterioplankton). In the heatmaps, yellow/beige colors are associated with a low relative abundance of the transcripts from the gene in a treatment, while dark red colors are associated with high relative abundance. Source data are provided as a Source Data file.

expressed between sterile black- and whitewater treatments (Fig. 4b). The expression of those nine genes (referred to as the "shared response" in Fig. 4) was independent from the presence/absence of bacteria. They included three monocarboxylic acid transporters (the solute carrier family 16 members 9a (SLC16A9A), the monocarboxylate transporters family 16 member 6b (SLC16A6B), and member 9b (SLC16A9B)), in addition to transporters of amino acids (the sodium-dependent neutral amino-acid transporter SLC1A4 and the L-amino-acid transmembrane transporter SLC43A2A), $Na^+$ (solute carrier family 20 member 1b (SLC20A1B)), $NH_4^+$ (the Rh family C glycoprotein B (RHCGB)), $Cl^-$ (the voltage-gated chloride ion channel CLCN5B) and $Ca^{2+}$ (the calcium voltage-gated channel subunit alpha 1a (CACNA1AA)). Previous studies focused on the physiological responses of zebrafish to acidic water also observed an overexpression of $Ca^{2+}$ channels[69] and Rh family C glycoproteins (RHCGs)[70]. RHCGs excrete ammonium from the gills and are potentially involved in $Na^+$ uptake via their interactions with the $Na^+/H^+$ exchangers[70].

Interestingly, a set of 22 differentially expressed genes between black- and whitewater treatments, were differentially expressed in non-sterile treatments but not in sterile treatments (Fig. 4a). They included several genes associated with $Ca^{2+}$ and $Na^+$ uptake systems (the transient receptor potential cation channel subfamily M member 4a (TRPM4A), the calcium-sodium antiporter SLC8A1A, the voltage-gated sodium channel SCN5LAB, the voltage-dependant calcium channels CACNA2D3A and CACNA2D2B). Numerous genes associated with transporters of $K^+$ were also included in this set (the sodium-chloride cotransporter SLC12A3, the potassium voltage-gated channel KCNH1B, the potassium channel KCNK3A and the nucleotide-gated potassium channel HCN2B). Analogously, results from two native Amazonian species (*S. rhombeus* and *T. albus*) show that the abundances of potassium transport proteins' transcripts were correlated to the expression of bacterial genes (the lysine/arginine/ornithine ABC transporter argT, the proton glutamate symport protein glutP/glutR, and the major outer membrane protein porB).

In addition to transporters of different amino acids such as glycine and ornithine (i.e., the solute carriers SLC7A8A, SLC7A4, SLC25A38A, SLC25A15B), other genes which expression was potentially influenced

by host-microbe interactions from this set include the heme transporter SLC48A1B and the $Na^+$/dicarboxylate symporter SLC1A8B. Heme transporter: The iron-containing cofactor heme is the most abundant source of iron for bacterial symbionts. Thus, host and bacterial heme homeostatic factors inter-mingle and compete for available heme, leading to the evolution of numerous systems for heme trafficking (reviewed in ref. 71). In our study, differences between the microbial community composition of black- and whitewater non-sterile treatments (Fig. 6e) could lead to different heme trafficking patterns and differential expression of the heme transporter. $Na^+$/dicarboxylate symporter: Dicarboxylates such as aspartate and malate are high-quality substrates for host nitrogen assimilation but are also important interkingdom signaling molecules that synchronize host and microbial physiological responses[72]. Interestingly, the amino-acid transporters, heme transporter, and $Na^+$/dicarboxylate symporter transcripts are among those that were more expressed in non-sterile blackwater (BWN) than in inverted non-sterile blackwater (BWI; blackwater inoculated with whitewater bacterioplankton), thus they could be related to the lower survival rates in this second treatment.

Finally, the response of zebrafish larvae to non-sterile blackwater also included the overexpression of immunity genes associated with cross-kingdom communication, such as the Toll-like receptor 13 (TLR13) and the ITGA2 integrin genes (in bold in Fig. 4). In this regard, we characterized the bacterial communities present in the non-sterile black- and whitewater treatments and showed that blackwater communities were characterized by a high relative abundance of Betaproteobacterial transcripts (87.7% in black- versus 14.9% in whitewater). A multi-level pattern analysis showed that all 77 biomarker ASVs were from the family Comamonadaceae. Transcripts from this family in non-sterile blackwater originated from *Acidovorax* (also found in fish from whitewater treatments, and identified as a microbiome contaminant in[56,73]) and *Diaphorobacter* (only found on non-sterile blackwater fish). These clades were not identified as blackwater fish gill biomarkers in any of the four native Amazonian fish considered, suggesting the presence of host-specific effects. These genera were not detected in bacterioplankton samples suggesting that, like the blackwater biomarkers of native Amazonian fish, they

show an important niche preference for fish. *Diaphorobacter* has been identified as one of the core members of the zebrafish microbiome[74]. It is unclear if this genus is commensal or beneficial for zebrafish in blackwater, as none of the bacterial genes correlated to Amazonian fish host responses to blackwater (identified in Fig. 3) were detected in this clade. However, a previous study on the skin mucus microbiome of a laboratory-raised Amazonian blackwater fish (*Symphysodon aequifasciata*) identified *Diaphorobacter* as a potential probiont[75]. Furthermore, the different taxonomy of the bacterial blackwater biomarkers identified between native Amazonian fish and zebrafish might also be affected by the age of the sampled specimens (adult fish were targeted on the field, while zebrafish larvae were used in the laboratory experiment).

### Perspective on DOC-microbiome-host interactions in blackwater

Evidence from different fish clades suggests that blackwater DOC could mitigate the ionoregulatory disturbances caused by acidic pH[16,17]. DOC molecules can bind directly to the surface of gill cells[20], but how exactly they interact with fish is still unresolved. Moreover, the protective roles of DOC are highly dependent on its chemical characteristics or integrity[25,26]. In aquatic ecosystems, bacteria are known to transform DOC in different ways[76]. For instance, bacterioplankton from Amazonian blackwaters possess the genomic potential to transform/metabolize some DOC components such as humic acids[50,77]. A recent study found that *Polynucleobacter* from the Amazon basin had genes involved in all the degradation steps of humic DOC[50]. Our study identified *Polynucleobacter* as one of the blackwater gill biomarkers in native Amazonian fish (Fig. 4). The potential interference of these bacteria with DOC at gill cell boundaries, as well as the potential interactions with DOC and overexpressed blackwater gill surface proteins such as integrins, merits further investigation.

Overall, this work reported the host-microbe responses of four Amazonian fish species to blackwater. We found that Amazonian fish transcriptional responses to blackwater were host species-specific. Our results challenged the current literature stating that Cichliformes rely on tight regulation of ion efflux and that Characiformes rely on ion transport systems that support high ion uptake: In three out of four native Amazonian species considered, both ionoregulatory strategies were observed. Furthermore, four of our main results suggest that host-microbe interactions are involved in the ionoregulatory response of fish-microbe systems to blackwater environments. First, in two native species and in zebrafish larvae, the response to blackwater

included the overexpression of immunity genes (i.e., Toll-like receptor and integrins) suggesting that interkingdom communication is involved in the response to blackwater. Second, blackwater fish gill microbiomes were significantly enriched in betaproteobacterial genera with the potential to interfere with gill epithelial permeability. In this regard, the blackwater-specific bacterium *Aquitalea* could potentially be involved in Amazonian fish response to blackwater via its potential to produce indole; or to express the bacterial porin porB, which interferes with the host membrane ion selectivity. Third, in *T. albus* and *S. rhombeus*, the expression of several potassium transport proteins was correlated to bacterial gene expression, a pattern that was also observed in the zebrafish laboratory experiment. Fourth, we showed that the normal response of zebrafish hosts to blackwater was mostly altered in the absence of microbiota. Furthermore, an interesting observation is the fact that the presence of whitewater bacterioplankton only conferred weak survival, stressing that only blackwater bacterioplankton strains provide key microbially-derived functions enabling optimal fish host survival in blackwater. Contrastingly, survival in whitewater was optimal in all treatments. In conclusion, our work suggests that using a more holistic approach encompassing both host and microbe contributions would benefit our understanding of the adaptations supporting life in one of Earth's most hostile and paradoxically most species-rich aquatic ecosystem.

## Methods
### Ethics approval
This study was carried out in accordance with the recommendations of the Ethics Committee for the Use of Animals of the Brazilian National Institute of Research of the Amazon (Manaus, Brazil) (permit # 29837-14) and the Animal Protection Committee of Laval University (Quebec, Canada) (permit # 2018021-1).

### Sample collection
**Field sampling.** Fish were collected from 15 sampling sites distributed along a hydrochemical gradient from the upper Brazilian Amazon in October-November 2018 and 2019 (dry season). GPS coordinates and a map of all sites are found in Supplementary Table 1 and Fig. 5, respectively. Four species were collected: Flag cichlids (*Mesonauta festivus*), freshwater sardines (*Triportheus albus*), peacock bass (*Cichla* spp.), and black piranhas (*Serrasalmus rhombeus*). These species were chosen for their large-scale distribution in the Amazonian basin, their relatively high abundance at every sampling site, and their tolerance to the

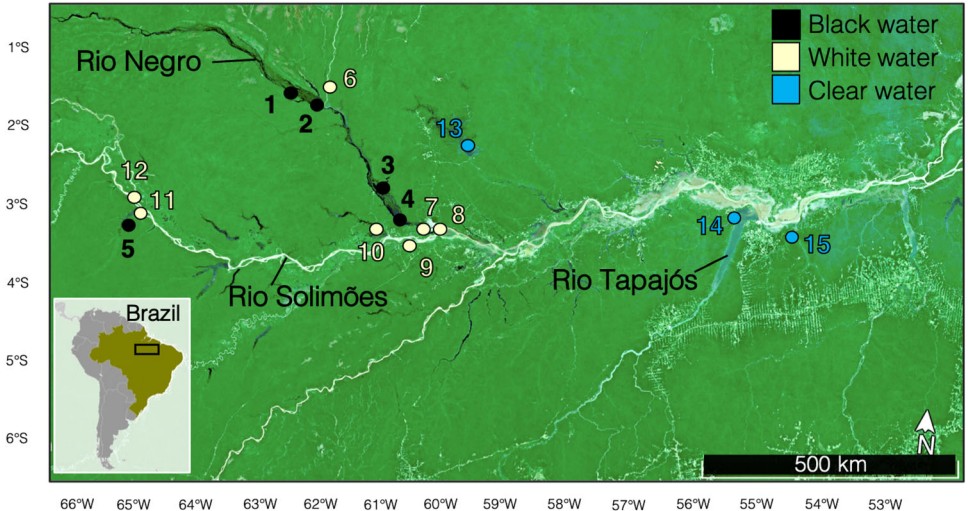

**Fig. 5 | Map of the sampling sites visited for this study.** Markers associated to every site are colored according to their water type (black for blackwater, beige for whitewater, and blue for clearwater samples).

contrasting environmental conditions (i.e., three Amazonian water types)[78]. This species assemblage represented two abundant teleosteans orders in the Amazon, the Cichliformes (*M. festivus* and *Cichla* spp.) and Characiformes (*S. rhombeus* and *T. albus*), and both orders were represented by an omnivorous and a carnivorous species. Of the omnivorous species, *M. festivus* feeds off the benthos and plant detritus[79], while *T. albus* feeds on fruits, seeds, and insects at the water surface[80]. Of the carnivorous species, *Cichla* spp. is piscivorous[81], while *S. rhombeus* will accept anything from fish to mussels[82]. All four species live on river and lake margins, are often found in large aggregations (except older individuals of *Cichla* spp. and *S. rhombeus*), and are not known to migrate between different habitats[8,78].

We aimed to collect up to twenty fish specimens per species per site (*N* target = 1200) using a combination of small seine net- and line fishing. A total of 964 specimens were successfully collected (number of specimens collected per site is in Supplementary Table 4). Specimens were measured and weighed (data in Supplementary file "Weight_length_table.xlsx"). One whole gill arch was randomly chosen and was sampled from each specimen immediately after collection using sterile dissection tools (EtOH 70%). The gill was sampled as it is the organ mostly responsible for ionoregulatory processes associated with the fish physiological response to different water types (reviewed in ref. [6]).

To consider environmental bacterioplankton activity in our analyses, we also sampled six water sample replicates per site at a depth of 30 cm in 2 L Nalgene™ bottles (cat #DS2205-0210, Thermo Fisher Scientific, Waltham (MA) USA). Filtration was performed as in[83] through 0.22 μm-pore size polyethersulfone Sterivex™ filters (cat #SVGP01050, Millipore, Burlington (MA), USA) less than 30 minutes after collection. Gill and filter samples were preserved in 2 mL of NAP conservation buffer to preserve RNA integrity[84] and then stored at −80 °C until processing.

We measured environmental variables (*n* = 34) associated to the Amazonian hydrochemical gradients[85] (Supplementary Tables 5–8). Methods used for these measurements are detailed in ref. [50]. Of the 15 sampling sites, five were blackwater, seven were whitewater and three were clearwater habitats (Supplementary Table 1). Whitewaters and clearwaters do not impose the ionoregulatory challenges characteristic of blackwaters: they typically have circumneutral pH, lower DOC concentration (~5 mg L$^{-1}$ in white- and clearwater but ~10-15 mg L$^{-1}$ in blackwater), and higher conductivity levels (~20–140 μS in white- and clearwater but <20 μS in blackwater)[86,87]. The turbid whitewaters are most common in the Andean Rio Solimões watershed and crystalline clearwaters are mostly found in the pre-Cambrian Rio Tapajos watershed[4,85]. The water types (Supplementary Table 1) were assigned from the measure of physicochemical parameters and confirmed the a priori knowledge of the water types of the sampled environments (Supplementary Tables 5–8).

**Laboratory experiment.** To isolate the fish physiological response to blackwater from potential bacterial interference, we performed a transplantation experiment with axenic zebrafish in a germ-free context. Zebrafish are non-native to Amazonia but provided a good model for our study since they are acid tolerant, extensive literature exists on this species and the production of sterile larvae is easily performed. Freshly fertilized eggs from wild type zebrafish were sterilized using the protocol of[88]. In brief, eggs were sterilized in three consecutive steps using a cocktail of three antibiotics (ampicillin, kanamycin, amphotericin B), polyvinylpyrrolidone–iodine 0.1%, and sodium hypochlorite (bleach) 0.003%. Then, they were transferred to sterile 160 mL (75 cm²) cell-culture flasks (Starstedt™, cat #101093-411) (40 larvae per flask) containing 50 mL of embryo medium solution prepared as in[89] and sterilized by filtration on 0.22 μM membranes (Nalgene™ Rapid-Flow™ vaccum filter, cat

#564-0020) and 1 h exposition to UV. The cell-culture flasks were incubated at 28.5 °C (VWR® B.O.D. refrigerated Peltier incubator, cat #89510-742) and after 24 h, dead eggs were removed (~30% of total). The eggs hatched in embryo medium after 48 h and larvae were then transferred to their respective experimental groups after 96 h. Water changes of 50% were done daily using embryo medium. The experiment ended before first feeding.

There were eight experimental groups, four containing blackwater from the Santo Alberto site (1°23'29.8"S 61°59'35.3"W) and four with whitewater from Lago des Pirates (3°15'19.2"S 64°41'44.3"W). The transport and storage (4 °C) of this water from the field to the lab had minor impacts on the bacterioplankton community (Supplementary Fig. 7) and values of physicochemical parameters (Supplementary Table 9). For each water type, there were four experimental conditions:[1] non-sterile water (i.e., with native bacterioplankton)[2], inverted non-sterile water (i.e., with bacterioplankton and sediments from the opposite water type)[3], sterile water (i.e., no bacterioplankton)[4], inverted sterile water (i.e., same as group #2 but sterile). The water was prepared identically for all experimental groups, until the last mixing step. First, water was aliquoted in 50 mL Falcon™ tubes (Thermo Fisher Scientific, cat #14-432-22) and centrifuged 15 min at 6000 g (4 °C). The pellets were separated from the supernatants in different tubes. All supernatants were autoclaved 20 min at 120 °C. Pellets from sterile groups were autoclaved identically. Then, the pellets were mixed with supernatants to produce the eight experimental groups, according to Fig. 6.

At 96 h post-fertilization, 100% of the embryo medium from cell-culture flasks was gradually changed for experimental water, over the course of 1 h of acclimation. Again, the cell-culture flasks were incubated at 28.5 °C for 18 h of exposition time. Preliminary tests showed that longer expositions times (i.e., 24 h) resulted in high mortality rates (~90%) of larvae in sterile blackwater groups. After 18 h, whole larvae were sacrificed by immersion in NAP buffer and were conserved in that medium at −80 °C. Before proceeding to RNA extraction, the larvae heads were isolated via dissection under microscope with sterile tools (EtOH 70%) and conserved in TRIzol™. The whole heads rather than the gills only were isolated since at that early developing stage, ionocytes of zebrafish larvae are scattered on the anterior region of the larvae and still migrating towards the underdeveloped gills[90]. Heads from ten larvae of the same experimental group were pooled together in each sample and there were two replicate samples per experimental group (*N* = 8 groups). The experiment was repeated two times, for a total of 320 larvae sampled, grouped in 16 pools of 20 larvae. Water sampling was also undertaken to examine bacterioplankton communities in the different treatments, as described in the "*Field sampling*" section. Water sampling was done at 96 h (right before exposing the larvae to the different treatments) and after the 18 h exposition. The sterility of larvae, embryo medium and water from sterile experimental groups was assessed first via aerobic culture on tryptic soy agar plates on which no colonies appeared after 14 days at 28.5 °C, then using 16 S rRNA gene cDNA sequencing from RNA extractions (Supplementary Fig. 8, 9).

## Sample processing

To remove NAP buffer from all biological samples, they were diluted by the addition of equal volumes of sterile ice-cold phosphate-buffered saline (PBS) and then centrifuged at 6000 × *g* for 15 min following the protocol of ref. [91]. The supernatant was discarded and the pellet was conserved in 1 mL of TRIzol™ (cat # 15596026, Thermo Fischer Scientific, Waltham (MA) USA). RNA extractions of Amazonian fish gills (unweighted whole gill arches), zebrafish, and bacterioplankton samples were performed according to the manufacturer's instructions of TRIzol™. The general measures taken to reduce the impact of potential contaminants during sample processing, according to recommendations in Eisenhofer et al.[92], are detailed in

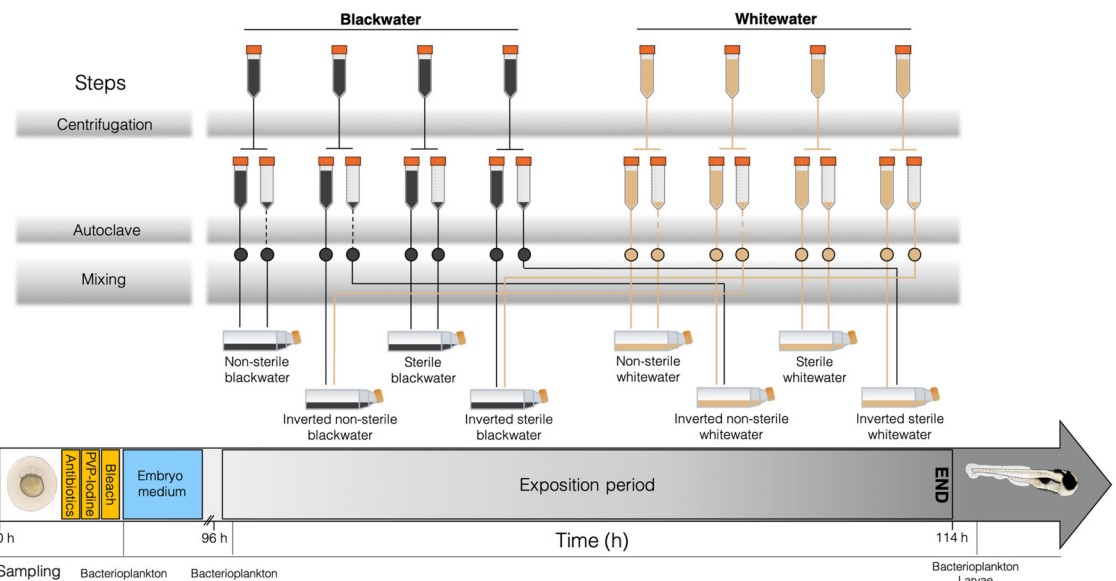

**Fig. 6 | Summary of the germ-free zebrafish experiment, including the main steps necessary to prepare the eight experimental conditions.** In the second row of Falcon™ tubes, full tubes represent supernatant while empty ones represent pellets (suspended material in original water, including sediments and bacterioplankton). Dashed lines represent steps that were not undertaken in the preparation of the referred groups.

Supplementary Methods. All biological materials (RNA extractions) are available from the authors upon request.

This study used two sequencing approaches[1]: Dual-RNASeq sequencing to study gene expression of fish and microbes (samples were pooled per species and sampling site to reduce costs)[2]; 16 S rRNA sequencing to characterize interindividual variability in terms of taxonomical distribution of transcriptomic activity among microbiome samples (samples not pooled).

### Dual-RNASeq libraries
Equal quantities of RNA from every sample were pooled by species and by sampling site for wild Amazonian fish gills (N = 55 pools). Samples were pooled per experimental group for zebrafish larvae samples. No dual-RNASeq was done for bacterioplankton samples. RNA integrity was assessed on Agilent BioanalyzerR RNA 6000 Nano/Pico Chips (cat # 5067-1511, Agilent Technologies Inc., Santa Clara (CA) USA). Depletion of rRNA was done using QIAseq FastSelect −5S/16 S/23 S (cat #334385, QIAGEN, Hilden, Germany) and QIAseq FastSelect Human Mouse Rat (cat #335921, QIAGEN, Hilden, Germany) kits according to the manufacturer's instructions. RNA fragmentation was done at 89 °C for 7 min. Then, NEBNext® Ultra II Directional RNA Library Prep Kit for Illumina® (cat # E7765S/L, New England Biolabs, Canada) was used for the synthesis of first and second strands cDNA, adaptor ligation (100-fold dilution of NEBNext Adaptor) and PCR enrichment of adaptor-ligated cDNA (13 PCR cycles) according to the manufacturer's instructions. Library quality was assessed on Agilent Bioanalyzer High Sensitivity DNA chip (cat # 5067-4626, Agilent Technologies Inc., Santa Clara (CA) USA). Sequencing was performed on two lanes of NovaSeq 6000 by the Next Generation Sequencing Platform of CHU de Québec research center.

For each host species, *salmon v1.3.0* (https://github.com/COMBINE-lab/salmon) was used in mapping mode to quantify the per-sample expression level. Raw reads were pseudo-aligned to the transcriptome of the closest available species with *salmon*. The transcriptome of the tilapia (RefSeq: GCF_001858045.2) was used for *M. festivus* and *Cichla* spp., the red-belly piranha transcriptome (RefSeq: GCF_015220715.1) was used for *S. rhombeus* and *T. albus*, and the zebrafish transcriptome (RefSeq: GCF_000002035.6) was used for *D. rerio*. The genomes of the same available species were specified as a decoy (salmon quant -l A −validateMappings) to obtain the Transcripts Per Million (TPM) values used for downstream analyses.

For each fish species, bacterial cDNA sequences were explored using *SqueezeMeta v1.5.1*[93] in co-assembly mode (*SqueezeMeta.ol -m co-assembly −cleaning*) following the simple *SqueezeMeta* pipeline. The results were then input to R (version 4.0.2) were analyzed using the *SqueezeMeta*-related R package *SQMtools*[94] which was used for downstream exploration of these datasets.

### 16 S rRNA gene libraries
Microbial community transcriptional activity was assessed on each sample (N = 964, in addition to three blank samples) using a 16 S rRNA approach conducted on RNA extracts as detailed in[50]. In brief: Retrotranscription was done with qScript cDNA synthesis kit (cat #95048-100) from QuantaBio (Beverly (MA), USA) according to the manufacturer's instructions. Then, the 16 S rRNA V3-V4 section (≈500 bp) was amplified using the forward and reverse primers 347 F (5'-GGAGGCAGCAGTRRGGAAT-3') and 803 R (5'-CTACCRGGGTATC-TAATCC-3')[95]. The first PCR of Amazonian fish gill and zebrafish samples was done according to the manufacturer's instructions of Q5® High-Fidelity DNA Polymerase from New England BioLabs® Inc (cat # M0491S) with an annealing temperature of 64 °C and 35 amplification cycles. The first PCR of bacterioplankton samples was done according to the manufacturer's instructions of the QIAGEN® Multiplex PCR kit (cat #206143, QIAGEN, Hilden Germany) using an annealing temperature of 60 °C and 30 amplification cycles - a higher sensitivity protocol adapted to their lower cDNA concentration. To add barcodes (indexes), a second PCR was done using Q5® High-Fidelity DNA Polymerase for gill, zebrafish, and bacterioplankton libraries (12 cycles). Amplified cDNA was purified with AMPure beads (cat #A63880, Beckman Coulter, Pasadena (CA), USA), according to the manufacturer's instructions. Post-PCR cDNA concentrations were assessed on Qubit™ (cat # Q33238, Thermo Fisher Scientific, Waltham (MA) USA) and after electrophoresis on 2% agarose gels. Multiplex sequencing was performed on Illumina MiSeq by the *Plateforme d'Analyses Génomiques* at the *Institut de Biologie Intégrative et des Systèmes de l'Université Laval*.

Amplicon sequence variant (ASV) picking was done with the R package *DADA2*[96] using the following commands: *filterAndTrim* for reads quality control (forward and reverse truncation lengths of 290

and 270; forward and reverse maximum expected error of 2 and 3), *learnErrors* for error rate learning, *derepFastq* for dereplication, *mergePairs* for merging, *DADA* for ASV inference and *removeBimeraDenovo* for chimera removal. ASVs were annotated using *blastn* matches against NCBI "16 S Microbial" database. Matches with >99% identity were assigned the reported taxonomic identity. Sequences below this threshold were annotated using a lowest common ancestor (LCA) method generated on the top 50 *blastn* matches obtained, as in *MEGAN*[97]. Potential cross contaminants inferred from sequenced blank samples were removed using *isContaminant* from the R *decontam* package[98] with the 0.4 default threshold (the relative abundance of those contaminants was represented in Supplementary Fig. 10). ASV tables, metadata files, and taxonomy information were incorporated into phyloseq-type objects[99] before downstream analyses.

### Statistical analysis

The Amazonian fish (host) physiological response to blackwater was assessed by differential gene expression (DGE) analysis. First, we built a database of genes known to be involved in ionoregulatory processes in fish facing acidic and ion-poor environments (see Supplementary Table 10) for each host species using available literature[6,31,49,100]. Then, to identify which of these genes were overexpressed in blackwater habitats, DGE analyses were computed on read counts tables of host RNA-Seq data using the R package *DESeq2*[101]. Differentially expressed genes had Wald test *p* values < 0.05 after Benjamini and Hochberg (BH) correction[102]. The relative abundance (in TPM) of genes overexpressed in blackwater habitats was then shown in heatmaps (Fig. 1).

The response of Amazonian fish gill microbial communities to blackwater was first assessed by computing distance-based redundancy analyses (dbRDA)[103] using *capscale* from the R package *vegan*[104]. The dbRDA analysis was based on Bray–Curtis distances[105] calculated from the 16 S rRNA gene dataset. Prior to computing dbRDA, multi-collinearity between explicative variables (environmental parameters) plotted in Fig. 2 was checked by measuring variance inflation factors (VIF) and by computing stepwise variable selection using *ordistep*[104]. Environmental parameters with VIF < 10 and selected by *ordistep* were kept as explicative variables. The effect of water type on gill microbiomes was measured using permutational analyses of variances (PERMANOVAs) with 999 permutations, again based on Bray–Curtis distances[105] calculated from the 16 S rRNA gene dataset. Then, the bacterial species that were the most strongly associated with blackwater fish gill microbiomes (referred as "blackwater biomarkers") were identified by multi-level pattern analysis using *multipatt* function from the R package *IndicSpecies*[106]. After 999 permutations of *multipatt*, ASVs with indicator value > 0.6 and *p* value < 0.01 for blackwater samples were kept as biomarkers for samples of that water type. The average indicator value of the genera represented by these biomarker ASVs, as well as their relative abundance (%) in different water types were respectively represented in bar- and boxplots (Fig. 2).

Correlations between the Amazonian fish host and microbial responses to blackwater were assessed at the functional level using Spearman correlations. Spearman correlations were calculated between the read counts of the differentially expressed host genes identified in Fig. 1 and those from all genes of the blackwater biomarkers identified on Fig. 2. Dual RNA-Seq data only was used to assess these correlations. Correlations with *p* value < 0.05 after Benjamini Hochberg correction were shown in scatterplots in Fig. 3. The relative abundance of the associated host and microbe dual RNA-Seq transcripts were shown in heatmaps with stacked barplots representing the taxonomic annotation of the bacterial taxa from which the transcripts originated (Fig. 3).

To understand how fish hosts respond to blackwater in the absence of gill bacterial symbionts, we conducted DGE on host transcriptomes from zebrafish larvae (dual-RNASeq data). The same gene database used for Amazonian fish hosts was used for zebrafish larvae

(i.e., genes potentially involved in the host response to blackwater; Supplementary Table 10). We used *DESeq2* to carry DGE analyses on the transcripts from these genes, between zebrafish from different water types (black- versus whitewater). Separate analyses were done for non-sterile (Fig. 4a, b) and sterile (Fig. 4c, d) groups. Differentially expressed genes had Wald test *p* values < 0.05 after BH correction[102]. The relative abundance (in TPM) of genes overexpressed in blackwater treatments was shown in heatmaps (Fig. 4a–d). To identify which bacteria might be associated with the different results observed between sterile and non-sterile DGE analyses of host transcripts, we first characterized the relative abundance of the bacterial classes in non-sterile black- and whitewater zebrafish larvae (using the 16 S rRNA gene dataset) (Fig. 4e). Then, blackwater biomarker ASVs were identified from this dataset using a multi-level pattern analysis (*multipatt*), as previously done for Amazonian fish gill microbiomes. The relative abundance of transcripts from these taxa in the dual RNA-Seq dataset, as well as their taxa of origin, were represented in stacked barplots (Fig. 4f).

### Reducing potential for contamination

The following approaches were used to reduce potential contamination:

- Randomization was conducted at several stages of the project:
- The different fish species were randomly sampled on the field (not all specimens from a single species were sampled at once).
- In the zebrafish experiment, fish from different treatments were randomly sampled (not all fish from the same treatment were sampled at once).
- To prevent day-to-day/batch effect, each extraction batch included randomly chosen samples from a variety of fish species. In addition, the samples from different species, sites, and experiments were randomly distributed on the PCR plates.
- A team of two researchers was simultaneously conducting RNA extractions (samples were randomly assigned to each researcher) and processing all samples using the same equipment.
- Both researchers were always wearing protective clothing (gloves, lab coats, and masks) and were working in a hood when possible.
- Reagents and all equipment were treated with UV before usage when possible.
- Library preparation was conducted in a separate room and using separate equipment from RNA extraction.
- Tips with filters were used exclusively to reduce potential cross-contamination.

### Ethics and inclusion statement

This project is a collaborative effort between scientists from Université Laval (Canada), Instituto Nacional de Pesquisas da Amazonia (Brazil), and LaTrobe University (Australia). The field work occurred in Brazil and the local collaborators (JC, DB, ALV) were involved in the study design, study implementation, and authorship of the publication. A capacity-building plan (international internship and training in microbial ecology) was developed for local collaborators. Relevant local research (e.g., Saudakas-Henrique et al. 2019, 2021; Duarte et al. 2016, 2017, 2022; Araújo et al. 2016) was taken into account in citations. Local research support systems were compensated fairly for their contributions.

### Reporting summary

Further information on research design is available in the Nature Portfolio Reporting Summary linked to this article.

## Data availability

The datasets generated for the current study can be found in the Sequence Read Archive (SRA) repository, BioProjectIDs: PRJNA839167, PRJNA839174, PRJNA902364, PRJNA902723, PRJNA901905, PRJNA902720, PRJNA902365, PRJNA902722, PRJNA902358 (data from

gills), PRJNA736442 and PRJNA736450 (data from bacterioplankton). The NCBI "16 S Microbial" database can be accessed here: https://www.ncbi.nlm.nih.gov/refseq/targetedloci/16S_process/. Source data for all figures are provided as a Source Data file. Source data are provided with this paper.

## Code availability

All metadata, input files, and R scripts used for data analysis are freely available on the Open Science Network platform (URL: https://osf.io/qea5j/).

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

## Acknowledgements

We thank the National Geographic Society (EC-50992R-18, FES), NSERC (FES), MITACS (IT10578, FES), and Ressources Aquatiques Québec (FES) for awarding travel and field-work grants. This study was part of the NSERC Discovery grant (6333) of ND, the INCT ADAPTA project of ALV, and supported by a Canada-Brazil Awards—Joint Research Project grant from Global Affairs Canada (CBJRP-2018-2020 CA-1, ND, and ALV), by CNPq, FAPEAM, and CAPES. We thank Thiago Nascimento, Reginaldo Oliveira, and Nazaré Paula for technical support with field work logisitcs. We thank Roxanne Dhommée for her support in the molecular biology laboratory work. We thank Camille and Charlotte Desrosiers for graciously providing original drawings for this manuscript.

## Author contributions

F.-É.S., A.L.V., and N.D. designed the study; F.-É.S., N.L., A.H., D.B., and N.D. performed field sampling; F.-É.S., N.L., P.L.M., and J.C. conducted RNA extractions and prepared metatranscriptomic and 16 S rRNA libraries; F.-É.S., S.B., E.N. conducted the bioinformatical and statistical analyses; F.-É.S. wrote the manuscript; all authors revised the manuscript.

## Competing interests

The authors declare no competing interests.
