## [Peer Review File · Nature Communications]

Important role of endogenous microbial symbionts of fish gills in the challenging but highly biodiverse Amazonian blackwatersREVIEWER COMMENTS

Reviewer #1 (Remarks to the Author):

The paper by Sylvain et al. investigates the eco physiological role of fish gills microbiota, as the most vital organ of these animals interacting with the surrounding water, in an extreme environmental setting. Apart from this, another strength of the paper is that the field findings have been validated with a model organism in the lab under experimental conditions. The protective role of DOC in the gills of fish from those habitats remains elusive and this paper is a novel and well-designed contribution to this.

The paper is exceptionally clear and presents and discusses all acquired results. The used methodology, from field to lab experiments to bioinformatics and statistical analysis is appropriate and complete. Despite the volume of results being high, the authors have provided truly comprehensive figures in the manuscript and leaving the right ones, more technical, for the supplementary material.

I am very glad I am among the first ones who had the chance to read such a novel contribution and excellent manuscript.

I have only the following minor comments/clarifications:

L. 568: Was always the same gill arch sampled, or any arch was randomly chosen?

L. 620-1: Autoclaving is known to alter the chemical composition of sea- and freshwater (e.g. if high concentrations of sugars exist, then they can be partially caramelized; this is why several microbiological media are autoclaved in 121 for 15 and not for 20 minutes). Is there any information on the DOC composition? DOC measurements before and after sterilization?

L. 636-7: I am not quite sure if sacrificing zebrafish larvae by immersing them in NAP buffer, please check and report.

Reviewer #2 (Remarks to the Author):

Title: "Fish-microbe systems in the hostile but highly biodiverse Amazonian blackwaters"

Here the authors evaluate the host transcriptome and host microbiome of the gills of four species of fish along a gradient within the Amazonian blackwaters. They show an increase in Toll-like receptors and integrins. They suggest or imply that the bacteria on the gill interfere with the gill permeability. Finally, they performed an exposure experiment to zebrafish and show that the microbes in the blackwater somehow protect the fish against stress.

Overall, I think this is a really interesting study and well designed. I appreciate how the authors integrated both microbiome and host transcriptome along with a follow up interventional type study to test the hypotheses generated. A few criticisms: 1) the authors don't really describe the rationale for choosing the fish species they choose or give any ecological background to the distributions of those fishes in the ecosystem etc, 2) the authors don't really give a lot of intro or context to other gill microbiome papers published (could also be in discussion), 3) order of presentation of results it's a little strange, 4) authors overly describe the transcriptomes in that much of the samples were pooled rather than from individuals. (you can't say you did 320 samples when you did 16 pools of 20 samples each... in that case you did 16 samples), and 5) the interventional trial on fish larvae might not have been the best choice in that at that stage typically fish have ionocytes primarily concentrated in the skin and thus osmoregulation occurs through the skin rather than gills. It's not until they are larger that osmoregulation occurs in the gills although I am not a fish physiologist by training.

RE the order of the results: the abstract states that the authors describe gill microbiome and gill

meta-transcriptome but the first result is the fish transcriptome followed by the microbiome.

Line 24: it sounds like Amazonia blackwaters are an extreme environment with little life but this sentence makes it sound like there are many organisms? It would be good to put this into context for aquatic ecosystems – what makes it diverse – is it more diverse than a kelp forest, a coral reef, ___? Do fish live here full time or do they just go in to feed and then come back out to cleaner waters?

Line 61: Can you clarify the 8% statement... is this just the number of species of fish that live there divided by total number of freshwater fish? How many different types of families or orders live there or are they all from the same species etc.

Line 98-110, I think the intro could benefit from a little more context on existing microbiome studies of fish gills. For instance of the fish chosen to study (4 species), were there any previous reports that they even had a gill microbiome. Why did you choose these fish?

Line 114: I suggest giving a more quantitative description of the hydrochemical gradient evaluated. The '964' gill samples seems somewhat arbitrary – authors could be a little more specific in the experimental design here.

Line 129 – 131: Please describe why these species were chosen

Figure 1 shows the transcriptomes of pooled fish across the gradients although the actual DGE are not from a regression but from a general grouping of black water vs clear water. The results here make it difficult to see if there are any consistencies across species. For instance, for both of the Cichliformes, there are no overlapping gene expressions. For Charactiformes, again there are no overlapping expression patterns although some general differences are with integrins and MAPK. The overall presentation of these results are a little confusing. Is there a reason for any particular order? In general the interpretation is that each fish responded differently regardless of order and therefore there may be no evolutionary connection to physiological adaptation?

Figure 2 shows the gill microbiomes across individual fishes. Its clear that there are multiple clusters for the white water group but its not clear this is discussed by the authors. Can you please elaborate on why there might be different clusters here? Also there are multiple clusters in blackwater for some species (T albus and S rhombeus) – please describe whats happening.

◇ One figure which is missing is all of the species together... is species a greater differentiator than water origin? The transcriptome results suggest that species effect is very large – is that also the case for microbes?

Line 385: Table 1, how do the authors distinguish between autochthonous and allochthonous here? Also, how many replication trials were done – are these the results from a single replication trial ? If yes, since this is a very important finding, authors should repeat the trial to have replicates and ensure that the high survival rate from blackwater non sterile autochthonous wasn't just some random fluke from that day. I don't think all the transcriptome work needs to be repeated – just the mortality assay.

Line 501-513: Overall this is a great study, but I do feel the authors could discuss these findings in regards to some of the other recent large scale gill microbiome studies

PMID: 29453266: gill mb driven by host on coral reefs

PMID: 26094624: potential negative impacts of sedimentation on the gill microbiome and gill health in fish

PMID: 36396943: lack of gill mb associated with pelagic lifestyles

PMID: 35285678: gill mb of fw fish and relation to osmotic stress and host transcriptional response

Line 692: Microbiome research is at a stage where three negative controls for a sample size approaching 1000 is no longer appropriate. It is clear that some level of contamination is consistently apparent across studies and thus including positive and negative controls for each batch of extraction is now expected. Why do you not include positive controls throughout the

process which is known to enable you to also distinguish between background contamination?
PMID: 30497919, PMID: 25387460, PMID: 31239396, PMID: 30046175

Line 692: If humic substances are so common in these environments including the gills themselves, it would be interesting to know if that had a negative impact on your extraction purity or PCR amplification efficiency?

Line 716: Running the Decontam package using only 3 negatives is pretty much pointless. I would recommend not running this package at all or reporting the number of ASVs removed as part of running this package. The package does not account for well to well contaminants thus it is very likely that by running this package, you end up losing a lot of true microbial signals rather than contaminants.

Line 750-752: What did you use for FDR here?, please report. If no FDR was used, then it should be implemented and recalculated.

Table 1

METHODS

Line 568: did you use the whole gill, part of the gill (tissue), or a swab...also did you weigh out the gill amount used into the extraction?

◇ This will impact your microbiome result PMID: 34758745

Line 577: Is this a type 22 um ? should be 0.22 um?

Line 594-648: Why do the authors use such young fish in the experiment? When fish are young most of the osmoregulation occurs in the skin as ionocytes are concentrated there rather than the gills. Please clarify rational for using such young fish in the experiment or not looking at the skin microbiome

Line 658-659: I understand its expensive, but you are loosing a lot of variation at the individual level when pooling samples. Did you do this either as a pilot or have other publications which can verify this is an appropriate approach?

RESPONSE TO REVIEWERS' CREPES

Reviewer #1

Comment 1: The paper by Sylvain et al. investigates the eco physiological role of fish gills microbiota, as the most vital organ of these animals interacting with the surrounding water, in an extreme environmental setting. Apart from this, another strength of the paper is that the field findings have been validated with a model organism in the lab under experimental conditions. The protective role of DOC in the gills of fish from those habitats remains elusive and this paper is a novel and well-designed contribution to this.

The paper is exceptionally clear and presents and discusses all acquired results. The used methodology, from field to lab experiments to bioinformatics and statistical analysis is appropriate and complete. Despite the volume of results being high, the authors have provided truly comprehensive figures in the manuscript and leaving the right ones, more technical.detailed, for the supplementary material.

I am very glad I am among the first ones who had the chance to read such a novel contribution and excellent manuscript.

Response: Thank you for this comment! We are glad you enjoyed reading our work.

I have only the following minor comments/clarifications:

Comment 2: L. 568: Was always the same gill arch sampled, or any arch was randomly chosen?

Response: The gill arch was randomly chosen on each specimen. This information (**in bold**) was added to the revised manuscript (lines 573-575):

*“One gill arch **was randomly chosen** and was sampled from each specimen immediately after collection using sterile dissection tools (EtOH 70%).”*

Comment 3: L. 620-1: Autoclaving is known to alter the chemical composition of sea- and freshwater (e.g. if high concentrations of sugars exist , then they can be partially caramelized; this is why several microbiological media are autoclaved in 121 for 15 and not for 20 minutes). Is there any information on the DOC composition? DOC measurements before and after sterilization?

Response: The reviewer raises an interesting point. Before conducting the gnotobiotic zebrafish experiment, we performed some preliminary tests and observed that the average DOC molecule sizes were slightly inferior after autoclaving than before (results were not recorded). Other studies have also shown that autoclaving DOC can impact its structure (Andersson *et al.* 2018). For this reason, we decided to autoclave the water from all the treatments in the gnotobiotic zebrafish experiment, including the water from sterile and non-sterile treatments, to ensure that there was no confounding effect between the DOC state (autoclaved versus not autoclaved) and the presence of bacteria. The pellets of non-sterile sediments and bacteria (isolated from the water prior to the autoclave) were reintroduced into the autoclaved water samples for non-sterile treatments, as illustrated on Fig. 6. Thus, all treatments from this experiment were equally exposed to autoclaved DOC.

Reference

Andersson, M.G.I., Catalán, N., Rahman, Z., Tranvik, L.J., & Lindström, E.S. Effects of sterilization on dissolved organic carbon (DOC) composition and bacterial utilization of DOC from lakes. *Aquat Microb Ecol* **82**,199-208 (2018).

Comment 4: L. 636-7: I am not quite sure if sacrificing zebrafish larvae by immersing them in NAP buffer, please check and report.

Response: Indeed, the zebrafish larvae were sacrificed by immersion in NAP buffer. This information (**in bold**) was added to the revised manuscript (lines 652-654):

“After 18h, whole larvae were sacrificed by immersion in NAP buffer and were conserved in that medium at -80°C.”

Reviewer #2

Comment 5: Here the authors evaluate the host transcriptome and host microbiome of the gills of four species of fish along a gradient within the Amazonia blackwaters. They show an increase in Toll-like receptors and integrins. They suggest or imply that the bacteria on the gill interfere with the gill permeability. Finally, they performed an exposure experiment to zebrafish and show that the microbes in the blackwater somehow protect the fish against stress.

Overall, I think this is a really interesting study and well designed. I appreciate how the authors integrated both microbiome and host transcriptome along with a follow up interventional type study to test the hypotheses generated.

Response: Thank you for this comment!

Comment 6: A few criticisms: 1) the authors don't really describe the rationale for choosing the fish species they choose or give any ecological background to the distributions of those fishes in the ecosystem etc,

Response: This information was previously in the Supplementary material, but has now been moved to the Methods of the revised manuscript (lines 558-568):

*“ These species were chosen for their large-scale distribution in the Amazonian basin, their relatively high abundance at every sampling site, and their tolerance to the contrasting environmental conditions (i.e. three Amazonian water types) (78). This species assemblage represented two abundant teleostean orders in the Amazon, the Cichliformes (*M. festivus* and *Cichla* spp.) and Characiformes (*S. rhombeus* and *T. albus*), and both orders were represented by an omnivorous and a carnivorous species. Of the omnivorous species, *M. festivus* feeds off the benthos and plant detritus (79), while *T. albus* feeds on fruits, seeds and insects at the water surface (80). Of the carnivorous species, *Cichla* spp. is piscivorous (81), while *S. rhombeus* will accept anything from fish to mussels (82). All four species live on river and lake margins, are often found in large aggregations (except older individuals of *Cichla* spp. and *S. rhombeus*), and are not known to migrate between different habitats (8, 78). ”*

Comment 7: 2) the authors don't really give a lot of intro or context to other gill microbiome papers published (could also be in discussion)

Response: Please see our detailed response to **Comment 21** below.

Comment 8: 3) order of presentation of results it's a little strange

Response: Please see our detailed response to **Comments 11 and 17** below.

Comment 9: 4) authors overly describe the transcriptomes in that much of the samples were pooled rather than from individuals. (you can't say you did 320 samples when you did 16 pools of 20 samples each... in that case you did 16 samples)

Response: The reviewer is right, while 320 zebrafish were sampled, only 16 pools can be considered for statistical analyses. To prevent any confusion, the number 320 was removed from the Abstract and the Introduction. In the Methods section, the following specification (**in bold**) was added (lines 659-660):

*“ The experiment was repeated two times, for a total of 320 larvae sampled, **grouped in 16 pools of 20 larvae.** ”*

Comment 10: 5) the interventional trial on fish larvae might not have been the best choice in that at that stage typically fish have ionocytes primarily concentrated in the skin and thus osmoregulation occurs through the skin rather than gills. Its not until they are larger that osmoregulation occurs in the gills although I am not a fish physiologist by training.

Response: Please see our detailed response to **Comment 28** below.

Comment 11: RE the order of the results: the abstract states that the authors describe gill microbiome and gill meta-transcriptome but the first result is the fish transcriptome followed by the microbiome.

Response: The results are stated in the Abstract in the same order as they appear in the Results section. The following words (**in bold**) were added in the revised Abstract to clarify the fact that the first result presented refers to the host transcriptional response and not to the microbial transcriptome (lines 27-36):

*“ [...] Here, we characterize the physiological response of 964 fish-microbe systems from four blackwater Teleost species along a natural hydrochemical gradient, using dual RNA-Seq and 16S rRNA of gill samples. We find that **host transcriptional** responses to blackwaters are species-specific, but occasionally include the overexpression of Toll-receptors and integrins associated to interkingdom communication. Blackwater gill microbiomes are characterized by a transcriptionally-active betaproteobacterial cluster potentially interfering with epithelial permeability. We explore further blackwater fish-microbe interactions by analyzing transcriptomes of axenic zebrafish larvae exposed to sterile, non-sterile and inverted (non-native bacterioplankton) blackwater. We find that axenic zebrafish survive poorly when exposed to sterile/inverted blackwater. Overall, our results suggest a critical role for endogenous symbionts in blackwater fish physiology. ”*

Comment 12: Line 24: it sounds like Amazonia blackwaters are an extreme environment with little life but this sentence makes it sound like there are many organisms? It would be good to put

this into context for aquatic ecosystems – what makes it diverse – is it more diverse than a kelp forest, a coral reef, ___? Do fish live here full time or do they just go in to feed and then come back out to cleaner waters?

Response: Indeed, Amazonian blackwaters are very diverse environments. The most recent study on this matter estimated the fish species richness of the Rio Negro basin (the largest blackwater basin in South America) to more than 1700 species (Beltrão *et al.* 2019), comprising 17 orders, 56 families, and 389 genera. This number is above the estimated 1620 fish species on the Great Barrier Reef (Richards and Day 2018). Although some occasional fish species are known to migrate yearly between different water types (Araujo-Lima & Ruffino 2004; Barthem *et al.* 2017), most blackwater species (including the four species sampled in our study) live their entire life in this habitat (Beltrão *et al.* 2019).

The following information (**in bold**) has been added to the revised Introduction (lines 59-63):

*“Overall, the chemistry of these blackwaters appears so extreme that they would most likely be toxic to most freshwater fish, yet the ichthyofauna inhabiting these habitats is one of the most biodiverse on Earth, **with an estimated richness of 1700 species (8), surpassing the estimated 1620 fish species on the Great Barrier Reef (9).**”*

The following information has been added to the description of the sampled species in the revised Methods (lines 566-568):

*“ All four species live on river and lake margins, are often found in large aggregations, and **are not known to migrate between different habitats (8, 78).** ”*

References

Araujo-Lima, C.A.R.M., & Ruffino, M. L. Migratory fishes of the Brazilian Amazon. In J. Carolsfield, B. Harvey, C. Ross, & A. Baer (Eds.), *Migratory fishes of South America: Biology, fisheries and conservation status* (pp. 62–75). Victoria, BC: World Fisheries Trust/World Bank/IDRC. (2004).

Barthem, R. B., *et al.* (2017). Goliath catfish spawning in the far western Amazon confirmed by the distribution of mature adults, drifting larvae and migrating juveniles. *Sci Rep*, **7** (2017).

Beltrão, H., Zuanon, J., Ferreira, E. Checklist of the ichthyofauna of the Rio Negro basin in the Brazilian Amazon. *Zookeys*. **17**, 881:53-89 (2019).

Richards, Z.T., Day, J.C. Biodiversity of the Great Barrier Reef-how adequately is it protected? *PeerJ*, **8**, 6:e4747 (2018).

Comment 13: Line 61: Can you clarify the 8% statement... is this just the number of species of fish that live there divided by total number of freshwater fish? How many different types of families or orders live there or are they all from the same species etc.

Response: The structure of this statement has changed in the revised manuscript. Please refer to our response to **Comment 12**.

Comment 14: Line 98-110, I think the intro could benefit from a little more context on existing microbiome studies of fish gills. For instance of the fish chosen to study (4 species), were there any previous reports that they even had a gill microbiome. Why did you choose these fish?

Response: In the revised manuscript, we made sure to include:

- Additional information about our choice of targeted fish species (see our detailed response to **Comment 6**).
- References and information from other existing large scale gill microbiome studies (see our detailed response to **Comment 21**).

Comment 15: Line 114: I suggest giving a more quantitative description of the hydrochemical gradient evaluated. The '964' gill samples seems somewhat arbitrary – authors could be a little more specific in the experimental design here.

Response: The sentence referred to in this comment has been modified to the following (lines 117-120):

“We first examined fish and microbial transcriptional activity from over 900 gill samples from four native Amazonian blackwater teleosts collected across a hydrochemical gradient of 15 sampling sites comprising natural blackwater and non-blackwater (white and clearwater) environments.”

We aimed to collect 20 specimens per species per site (20 specimens x 4 species x 15 sites = 1200 targeted specimens). This information about the experimental design was added to the Methods section (lines 570-572):

“We aimed to collect up to twenty fish specimens per species per site (N target = 1200) using a combination of small seine net- and line fishing. A total of 964 specimens were successfully collected (number of specimens collected per site is in Suppl. Table 4).”

Comment 16: Line 129 – 131: Please describe why these species were chosen

Response: See our detailed response to **Comment 6**. This information has been added to the Methods of the revised manuscript (lines 558-561):

“These species were chosen for their large-scale distribution in the Amazonian basin, their relatively high abundance at every sampling site, and their tolerance to the contrasting environmental conditions (i.e. three Amazonian water types) (78).”

Comment 17: Figure 1 shows the transcriptomes of pooled fish across the gradients although the actual DGE are not from a regression but from a general grouping of black water vs clear water. The results here make it difficult to see if there are any consistencies across species. For instance, for both of the Cichliformes, there are no overlapping gene expressions. For Charactiformes, again there are no overlapping expression patterns although some general differences are with integrins and MAPK. The overall presentation of these results are a little confusing. Is there a reason for any particular order? In general the interpretation is that each fish responded differently regardless of order and therefore there may be no evolutionary connection to physiological adaptation?

Response: Based on the above comment from the reviewer, we noticed that *Cichla spp.* and *S. rombeus* were originally inverted (classified in the wrong order) on Fig. 1, which caused a mismatch with the corresponding text in the Results section of the previous manuscript version. This mistake was corrected in the revised manuscript: The revised Fig. 1 now presents the

differential gene expression results for each species within the two orders (both Cichliformes to the left and both Characiformes to the right). This follows the same order as in the “Results and discussion” section “**Amazonian fish host physiological response to blackwater**”.

The reviewer is also right when mentioning that: “*In general the interpretation is that each fish responded differently regardless of order and therefore there may be no evolutionary connection to physiological adaptation [...]*”.

The following information has been included in the corresponding Results section of the revised manuscript (lines 235-240):

“Overall, the results from these four species (Fig. 1) do not support the evolutionary convergence of physiological adaptation to blackwater, since each species responded differently to blackwater regardless of its order (with the exception of the overexpression of GAP junction proteins in both Cichliformes). Furthermore, the results challenge the current literature (15) stating that Cichliformes rely on tight regulation of ion efflux while Characiformes rely on high affinity ion uptake systems as both ionoregulatory strategies were observed on three species.”

Comment 18: Figure 2 shows the gill microbiomes across individual fishes. Its clear that there are multiple clusters for the white water group but its not clear this is discussed by the authors. Can you please elaborate on why there might be different clusters here? Also there are multiple clusters in blackwater for some species (T albus and S rhombeus) – please describe whats happening.

Response: The reviewer makes a good point: Samples from the same water type sometimes gather in more than one cluster, which comes from the effect of the “sampling site” factor. To document the effect of this factor, we added a new figure to the Supplementary Material of the revised manuscript, where the samples represented on the same distance-based redundancy analysis plots as Fig. 2 are colored by their sampling site of origin rather than by their water type.

The following figure (and the discussion of its results in the caption) has been added to the Supplementary Material of the revised manuscript:

Suppl. Fig. 5: Bray-Curtis distance-based redundancy analyses (dbRDA) on the gill bacterial (16S rRNA) community samples of (a) flag cichlid, (b) peacock bass, (c) sardine, and (d) black piranha. Each data point in the dbRDAs represents a microbiome sample. The samples are colored according to their sampling site of origin and their shape corresponds to the water type of the site sampled. The environmental parameters represented in the dbRDAs were selected by ordistep and had a VIF < 10. PERMANOVA results (999 permutations) exploring the differences in community composition between sampling sites are at the upper-right corner of each plot. “***” stands for PERMANOVA p-value < 0.001. df.res stands for residual degrees of freedom. Overall, the results show that the sampling site factor is significantly associated to microbiome composition (PERMANOVA p-values < 0.001) for each host species. Among other factors, differences in the physicochemical parameters between sites of the same water type might partly explain the effect of the sampling site factor (e.g. conductivity, pH, and concentrations of Al, Fe and Zn could explain differences between different clusters of whitewater sites, Suppl. Tables 7 and 8). To prevent confounding the water type and sampling site effects, fish were sampled from a minimum of three sampling sites per water type.

Finally, the following text was added to the Results and discussion section (lines 271-278):

“In this case, changes in gill bacterial communities according to water types could be associated to an interaction between host-associated and free-living bacterioplanktonic communities specific to each water type (as documented in (50)) or they could also occur in direct response to physicochemical parameters variations. These two environment-associated factors might also explain the significant differences in the gill microbiome from fish of different sampling sites (pval < 0.001, Suppl. Fig. 5), supporting previous studies that revealed the sensitivity of gill microbial assemblages to environmental factors (e.g. levels of sediments, salinity levels) (35, 53).”

Comment 19: One figure which is missing is all of the species together... is species a greater differentiator than water origin? The transcriptome results suggest that species effect is very large – is that also the case for microbes?

Response: We added a new figure (distance-based redundancy analysis) combining all species from all water types together in the revised Supplementary material (Suppl. Fig. 3). We also added the results of PERMANOVA tests to this figure. The results show that both water type and host species are significantly associated to microbiome composition (PERMANOVA p-values < 0.001 for both factors), however the water type factor is approximately 3x stronger than the host species factor (F water type = 22.2 versus F host species = 7.4). The clustering of samples by water type on the redundancy analysis plot is also much stronger (in Suppl. Fig. 3a) than clustering by host species (in Suppl. Fig. 3b). See new Suppl. Fig. 3 here below:

Suppl. Fig. 3: Bray-Curtis distance-based redundancy analyses with all samples together (all species and water types). Samples are colored according to their water type in (a) and to their host species in (b). Permutational analysis of variance (PERMANOVA, 999 permutations) results for each factor are at the bottom left of the corresponding plots. *** means p-value < 0.001. df.res means the residual degrees of freedom. Overall, the results show that both water type and host species are significantly associated to microbiome composition (PERMANOVA p-values < 0.001 for both factors), however the water type factor is approximately 3x stronger than the host species factor (F water type = 22.2 versus F host species = 7.4).

Comment 20: Line 385: Table 1, how do the authors distinguish between autochthonous and allochthonous here? Also, how many replication trials were done – are these the results from a

single replication trial? If yes, since this is a very important finding, authors should repeat the trial to have replicates and ensure that the high survival rate from blackwater non sterile autochthonous wasn't just some random fluke from that day. I don't think all the transcriptome work needs to be repeated – just the mortality assay.

Response: The following information was added to the caption of Table 1 to clarify the distinction between autochthonous and allochthonous bacterioplankton (lines 395-398):

“Autochthonous bacterioplankton refers to bacterioplankton from the same water type (e.g. whitewater bacterioplankton in whitewater), while allochthonous bacterioplankton refers to bacterioplankton from the opposite water type (e.g. whitewater bacterioplankton in blackwater).”

The experiment was repeated two times and there were two replicates of each treatment per experiment. The mortality percentages shown in Table 1 correspond to the mean mortality observed in these four observations per treatment. The standard deviation calculated on those four observations was added to Table 1. The following information was added to the caption of Table 1 (lines 399-400):

“Mortality rates are mean values and their standard deviation from the two replicates of the two times the experiment was conducted (four observations).”

Comment 21: Line 501-513: Overall this is a great study, but I do feel the authors could discuss these findings in regards to some of the other recent large scale gill microbiome studies

PMID: 29453266: gill mb driven by host on coral reefs

PMID: 26094624: potential negative impacts of sedimentation on the gill microbiome and gill health in fish

PMID: 36396943: lack of gill mb associated with pelagic lifestyles

PMID: 35285678: gill mb of fw fish and relation to osmotic stress and host transcriptional response

Response: We agree that it is important to introduce and discuss our findings with respect to other large scale gill microbiome studies, and we thank the review for suggesting some very interesting literature on this matter.

The following information, and references to the work of Pratte *et al.* (2018) and Minich *et al.* (2022), were added to the Results and discussion section (lines 255-263):

*“Bray-Curtis distance-based redundancy analyses (dbRDA) showed that gill microbiota samples of each host species clustered according to their water type of origin (species-specific analyses on Fig. 2, all species combined in Suppl. Fig. 3). Permutational analyses of variance (PERMANOVA) confirmed that the gill bacterial communities were significantly different between host species ($pval < 0.001$). A significant effect of host-specific factors on gill bacterial assemblages has been previously detected on large-scale investigations on saltwater fish by Pratte *et al.* (2018) (51) and most recently in a study by Minich *et al.* (2022) (52) which encompassed 101 species. This last study (52) also detected a significant correlation between alpha diversity measures and fish weight or trophic level, however we did not detect both of these correlations in our study (Suppl. Fig. 4).”*

Suppl. Fig. 4 was also added to the revised Supplementary Material:

Suppl. Fig. 4: Shannon diversity measure of gill microbiome samples from the four host species collected. “CAR” stands for carnivorous; “OMN” for omnivorous; “CHA” for Characiformes; “CIC” for Cichliformes. Gill microbiome Shannon diversity was not significantly different between host species, trophic levels or host phylogeny (Order) and was not significantly correlated to mean fish weight (g).

The following information (**in bold**) and references to the work of Hess *et al.* (2015) and Lai *et al.* (2022) were also added to the revised manuscript (lines 271-278):

*“In this case, changes in gill bacterial communities according to water types could be associated to an interaction between host-associated and free-living bacterioplanktonic communities specific to each water type (as documented in (50)) or they could also occur in direct response to physicochemical parameters variations. **These two environment-associated factors might also explain the significant differences in the gill microbiome from fish of different sampling sites (pval < 0.001, Suppl. Fig. 5), supporting previous studies that revealed the sensitivity of gill microbial assemblages to environmental factors (e.g. levels of sediments, salinity levels) (35, 53).”***

The following sentence and reference to the work of Lai *et al.* (2022) has been added to the revised Introduction (lines 109-114):

“A recent study on the medaka fish model (35), integrating transcriptomics and metagenomics to characterize the response of fish holobionts to hypotonic stress, suggested that host-bacterium interactions could facilitate the freshwater acclimation through the glycosaminoglycan and chitin pathways, but the functions involved remain

unknown, as the study inferred functional profiles with PICRUSt2 and not through a dual RNA-Seq approach.

The following reference to Lai *et al.* (2022) has also been added to the revised Results and discussion (lines 206-207):

“Previous studies have detected an overexpression of integrins in fish exposed to hypotonic environments (35, 44).”

References

Hess, S., Wenger, A. S., Ainsworth, T. D., & Rummer, J. L. Exposure of clownfish larvae to suspended sediment levels found on the Great Barrier Reef: Impacts on gill structure and microbiome. *Sci Rep*, **5**, 10561 (2015).

Lai, K. P., *et al.* Integrated omics approaches revealed the osmotic stress-responsive genes and microbiota in gill of marine medaka. *mSystems*, **7(2)**, e0004722 (2022).

Minich, J. J., *et al.* Host biology, ecology and the environment influence microbial biomass and diversity in 101 marine fish species. *Nat Comm*, **13(1)**, 6978 (2022).

Pratte, Z. A., Besson, M., Hollman, R. D., & Stewart, F. J. The gills of reef fish support a distinct microbiome influenced by host-specific factors. *Appl and Env Microbiol*, **84(9)**, e00063-18 (2018).

Comment 22: Line 692: Microbiome research is at a stage where three negative controls for a sample size approaching 1000 is no longer appropriate. It is clear that some level of contamination is consistently apparent across studies and thus including positive and negative controls for each batch of extraction is now expected. Why do you not include positive controls throughout the process which is known to enable you to also distinguish between background contamination? PMID: 30497919, PMID: 25387460, PMID: 31239396, PMID: 30046175

Response: The reviewer brings up an important point here concerning the control of contamination. We agree that limiting the presence of contamination and characterizing its extent are key to ensure that the biological signal detected is valid.

Controlling for potential contaminants is especially important for studies using samples with low biomass (e.g. ice, air, rocks, etc.) where contaminant DNA can dominate the biological signal (Eisenhofer *et al.* 2018). The main focus of our study was on whole gill samples (approximately 1 to 2 cm long) which, in comparison with the aforementioned substrates, should not be considered as “low biomass” samples. Following RNA extractions using TRIzol™, average RNA concentrations varied between 1000-1500 ng/μl. Thus, we should expect the relative abundance of contaminants to be minimal in such RNA-rich samples.

Nevertheless, to prevent and address the presence of potential contaminant sequences we put in place several measures in accordance to the ‘**R I D E**’ checklist (detailed in Eisenhofer *et al.* 2018 – PIMD 30497919):

R → Report the experimental design and approaches used to reduce and assess the contributions of contamination.

The following approaches were used to reduce potential contamination:

- Randomization was conducted at several stages of the project:
 - The different fish species were randomly sampled on the field (not all specimens from a single species were sampled at once).
 - In the zebrafish experiment, fish from different treatments were randomly sampled (not all fish from the same treatment were sampled at once).
 - To prevent day-to-day/batch effect, each extraction batch included randomly chosen samples from a variety of fish species. In addition, the samples from different species, sites and experiments were randomly distributed on the PCR plates.
- A team of two researchers were simultaneously conducting RNA extractions (samples were randomly assigned to each researcher) and processed all samples using the same equipment.
- Both researchers were always wearing protective clothing (gloves, lab coats and masks) and were working in a hood when possible.
- Reagents and all equipment were treated with UV before usage when possible.
- Library preparation was conducted in a separate room and using separate equipment than from RNA extraction.
- Tips with filters were used exclusively to reduce potential cross-contamination.

The aforementioned points have been added to Suppl. Methods section “*Reducing potential for contamination*”, as specified in the revised manuscript (lines 674-676):

“The general measures taken to reduce the impact of potential contaminants during sample processing, according to recommendations in Eisenhofer et al. 2018, are detailed in Suppl. Methods.”

The approaches used to assess the contribution of contamination in the dataset are detailed in the following subsections.

I → Include controls to assess contaminant DNA.

- The paper from Eisenhofer *et al.* (2018) details three types of blanks: Sampling blanks, extraction blanks and amplification blanks. Our study used sampling and amplification blanks. First, three sampling blanks (consisting of NAP conservation buffer without any biological sample) were extracted, amplified and sequenced. Since they were processed exactly in the same way as “normal” samples, these blanks accounted for potential contaminants in all steps of the sample processing. In addition, we added two amplification blanks (wells containing PCR reactants without any DNA template) on each plate (16 plates). The DNA from these blanks was undetectable on 2% agarose gels and after quantification on Qbit (concentration approx. 0 ng/μL). The amplification blanks were not sequenced but their extremely low DNA concentrations in comparison to those from gill samples suggested that the abundance of potential amplification contaminants was minor.

D → Determine the level of contamination by comparing biological samples to controls.

After sequencing of the blanks, we detected a total of 143 ASVs. The following genera were detected: *Micrococcus*, *Corynebacterium*, *Bacillus* and *Streptomyces*. Some of these genera represented commonly encountered contaminants in microbiome studies (see Table 1 in Eisenhofer *et al.* 2018), for instance *Corynebacterium* and *Bacillus* were detected in Salter *et al.* (2014), Glassing *et al.* (2016) and Weyrich *et al.* 2019.

Of these, 61 ASVs were identified as potential contaminants by the R package *decontam*. These ASVs had a low relative abundance in other samples (average relative abundance of 0.007 %, maximum of 1.54 %) and thus their removal from the dataset did not have an important impact on the observed gill microbial assemblages. The relative abundance of contaminants in “true” samples is shown on Suppl. Fig. 10, a new figure added to the revised Supplementary Material:

Suppl. Fig. 10: Barplot showing the relative abundance (in %) of the ASVs identified by potential contaminants ($N = 61$ ASVs) by the R package *decontam* (Davis *et al.* 2018). Negative controls and sterile samples were excluded from this analysis. The ASVs identified as contaminants had an average relative abundance of 0.007% and reached a maximum of 1.54%.

E → Explore contaminant taxa and report their impact on the interpretation of biological samples.

The Table 1 from the review of Eisenhofer *et al.* (2018) lists the taxa detected as contaminants in two or more studies from the literature. In the table below, we listed the bacterial genera “driving the signal” (either through the biomarkers analyses or the correlation analysis) for the blackwater fish in our study. Then, we checked whether these taxa were also found in the table of “potential contaminants” in the Eisenhofer *et al.* (2018) paper. Overall, of the 18 genera discussed in our study, only four were listed as potential contaminants in the literature. Furthermore, none of these 18 taxa were identified as contaminants in the negative controls or through the *decontam* analysis.

Table 1: List of the bacterial genera associated to the response to blackwater in our study and occurrence of these taxa as contaminants in other studies.

Genus associated to the response to blackwater	Experiment	Identified in	Potential contaminant? (Eisenhofer et al. 2018)	Also identified in
Aquitalea	Field	Biomarkers analysis (Figure 2)	No	
Chitinomonas	Field	Biomarkers analysis (Figure 2)	No	
Polynucleobacter	Field	Biomarkers analysis (Figure 2)	No	
Undibacterium	Field	Biomarkers analysis (Figure 2)	No	
Desulfovibrio	Field	Biomarkers analysis (Figure 2)	No	
Rhodobacter	Field	Biomarkers analysis (Figure 2)	No	
Acinetobacter	Field	Biomarkers analysis (Figure 2)	Yes	Tanner et al. (1998) Barton et al. (2006) Salter et al. (2014) Lauder et al. (2016) Weyrich et al. (2018)
Kinneretia	Field	Biomarkers analysis (Figure 2)	No	
Methylocaldum	Field	Biomarkers analysis (Figure 2)	No	
Neorickettsia	Field	Biomarkers analysis (Figure 2)	No	
Novosphingobium	Field	Biomarkers analysis (Figure 2)	Yes	Salter et al. (2014) Weyrich et al. (2018)
Vogesella	Field	Correlation analysis (Figure 3)	No	
Chryseobacterium	Field	Correlation analysis (Figure 3)	Yes	Salter et al. (2014) Weyrich et al. (2018)
Chromobacterium	Field	Correlation analysis (Figure 3)	No	
Acidovorax	Laboratory	Biomarkers analysis (Figure 4)	Yes	Barton et al. (2006) Salter et al. (2014)
Diaphorobacter	Laboratory	Biomarkers analysis (Figure 4)	No	
Variovorax	Laboratory	Biomarkers analysis (Figure 4)	No	
Ottowia	Laboratory	Biomarkers analysis (Figure 4)	No	

To disclose this information to the readers of the revised manuscript, we added the following statements (**in bold**) at lines (303-305):

“Of note, *Acinetobacter* and *Novosphingobium* were identified as potential contaminants of microbiome studies in the literature (56, 57), thus the results pertaining to these taxa should be interpreted with caution.”

And lines 364-367:

“Finally, we observed a correlation between the host expression of a Na⁺/H⁺ exchanger by *M. festivus*, and the bacterial expression of *atpF*, a bacterial H⁺-ATPase expressed by several taxa including *Undibacterium*, *Chryseobacterium* (identified as a potential microbiome contaminant in (56, 57)**), in addition to unidentified *Neisseriales* and *Burkholderiales* (Fig. 3c).”**

And lines 478-481:

“Transcripts from this family in non-sterile blackwater originated from *Acidovorax* (also found in fish from whitewater treatments, **and identified as a microbiome contaminant in (56, 73)) and *Diaphorobacter* (only found on non-sterile blackwater fish).”**

References

Barton, H. A., Taylor, N. M., Lubbers, B. R., & Pemberton, A. C. DNA extraction from low-biomass carbonate rock: An improved method with reduced contamination and the low-biomass contaminant database. *J of Microbiol Meth*, **66(1)**, 21–31 (2006).

Davis, N. M., Proctor, D. M., Holmes, S. P., Relman, D. A., & Callahan, B. J. Simple statistical identification and removal of contaminant sequences in marker-gene and metagenomics data. *Microbiome*, **6(1)**, 226 (2018).

Eisenhofer, R., Minich, J. J., Marotz, C., Cooper, A., Knight, R., & Weyrich, L. S. Contamination in low microbial biomass microbiome studies: Issues and recommendations. *Trends in Microbiol*, **27(2)**, 105–117 (2019).

Glassing, A., Dowd, S. E., Galandiuk, S., Davis, B., & Chiodini, R. J. Inherent bacterial DNA contamination of extraction and sequencing reagents may affect interpretation of microbiota in low bacterial biomass samples. *Gut Pathogens*, **8(24)** (2016).

Lauder, A. P., *et al.* Comparison of placenta samples with contamination controls does not provide evidence for a distinct placenta microbiota. *Microbiome*, **4(1)**, 29 (2016).

Salter, S. J., *et al.* Reagent and laboratory contamination can critically impact sequence-based microbiome analyses. *BMC Biol*, **12(87)** (2014).

Tanner, M. A., Goebel, B. M., Dojka, M. A., & Pace, N. R. Specific ribosomal DNA sequences from diverse environmental settings correlate with experimental contaminants. *Appl and Env Microbiol*, **64(8)**, 3110–3113 (1998).

Weyrich, L. S., *et al.* Laboratory contamination over time during low-biomass sample analysis. *Mol Ecol Res*, **19(4)**, 982–996 (2019).

Comment 23: Line 692: If humic substances are so common in these environments including the gills themselves, it would be interesting to know if that had a negative impact on your extraction purity or PCR amplification efficiency?

Response: The reviewer is right: Some studies observed that humic substances can reduce the yield of nucleic acid extractions (Wnuk *et al.* 2020). In our case, we did not observe any difference in RNA extractions yields, purity or PCR amplification efficiency between gill samples from humic-rich (blackwater) and humic-poor (white/clearwater) environments. Following RNA extractions using TRIzol™, most samples from all water types had RNA concentrations between 1000-1500 ng/μL. The absence of effect of humic substances on these samples could be due to their low concentrations on fish gills, or to the storage of gill samples in NAP buffer which, in some way, “rinsed” the gills and diluted any humic substances in the buffer which was discarded before RNA extraction.

Reference

Wnuk, E., *et al.* The effects of humic substances on DNA isolation from soils. *PeerJ*, **8**, e9378 (2020).

Comment 24: Line 716: Running the Decontam package using only 3 negatives is pretty much pointless. I would recommend not running this package at all or reporting the number of ASVs removed as part of running this package. The package does not account for well to well contaminants thus it is very likely that by running this package, you end up losing a lot of true microbial signals rather than contaminants.

Response: Please see our response to the **Comment 22** concerning the control of potential contaminants. The ASVs removed using this package (N = 61) only represented a minor fraction (0.07%) of all the ASVs for this dataset (N = 90,649). They also represented a low relative abundance of the total community (average of 0.007%, maximum of 1.54% per sample). None of these ASVs are from the genera “driving the blackwater microbiome response” (i.e. blackwater biomarkers, or correlated to the overexpression of host genes in blackwater).

As mentioned in response to a previous comment, a new figure has been added in Supplementary Material (Suppl. Fig. 10), to represent the relative abundance of potential contaminants in all samples. The following figure caption has also been added:

“Suppl. Fig. 10: Barplot showing the relative abundance (in %) of the ASVs identified as potential contaminants (N = 61 ASVs) by the R package decontam (Davis et al. 2018). Negative controls and sterile samples were excluded from this analysis. The ASVs identified as contaminants had an average relative abundance of 0.007% and reached a maximum of 1.54% per sample.”

The reviewer also makes a valid point: *Decontam* does not account for well-to-well contamination. However, we limited well-to-well effects by randomizing all samples across plates, and by using a manual single-tube extraction method rather than conducting the extractions on a machine. These good practices concur with some of the recommendations detailed in Minich *et al.* (2019).

Reference

Minich, J.J., Sanders, J.G., Amir, A., Humphrey, G., Gilbert, J.A., Knight, R. Quantifying and understanding well-to-well contamination in microbiome research. *mSystems*, **25;4(4)**, e00186-19 (2019).

Comment 25: Line 750-752: What did you use for FDR here?, please report. If no FDR was used, then it should be implemented and recalculated.

Response: The Benjamini and Hochberg (BH) correction was used as FDR. The following was clarified (see **bold** text) in the revised manuscript (lines 775-779):

*“Spearman correlations were calculated between the read counts of the differentially expressed host genes identified in Fig. 1 and those from all genes of the blackwater biomarkers identified on Fig. 2. Dual RNA-Seq data only was used to assess these correlations. Correlations with p-value < 0.05 after **Benjamini and Hochberg correction** were shown in scatter plots in Fig. 3.”*

Comment 26: Line 568: did you use the whole gill, part of the gill (tissue), or a swab...also did you weigh out the gill amount used into the extraction? This will impact your microbiome result PMID: 34758745

Response: We sampled whole gills (one randomly chosen whole gill arch per specimen). The following information (**in bold**) was added to the Methods (lines 573-575):

*“One **whole** gill arch was randomly chosen and was sampled from each specimen immediately after collection [...]”*

And lines 672-674:

*“RNA extractions of Amazonian fish gills (**unweighted whole gill arches**), [...] were performed according to the manufacturer’s instructions of TRIzol™.”*

Comment 27: Line 577: Is this a type 22 um ? should be 0.22 um?

Response: Indeed, it should be 0.22 µm. This was corrected in the revised manuscript (lines 582-583):

“Filtration was performed as in (70) through 0.22 µm-pore size polyethersulfone Sterivex™ filters [...]”

Comment 28: Line 594-648: Why do the authors use such young fish in the experiment? When fish are young most of the osmoregulation occurs in the skin as ionocytes are concentrated there rather than the gills. Please clarify rational for using such young fish in the experiment or not looking at the skin microbiome

Response: We absolutely agree that it would have been interesting to study the ionoregulation from older/mature zebrafish. However, this experiment had several biological and logistical constraints that forced us to use larvae rather than mature individuals:

(1) The sterilization protocol required to expose the zebrafish to polyvinylpyrrolidone–iodine 0.1% and to sodium hypochlorite (bleach) 0.003%. While the zebrafish eggs can withstand this treatment on the surface of their thick membrane, the delicate tissues of larval or juvenile fish would not be as resistant and the fish would most probably die rapidly following exposition to these concentrations. Thus, the sterilization protocol required us to sterilize eggs rather than larvae/juvenile fish.

(2) Literature showed that zebrafish larvae start to eat external food after five days post fertilization (Westerfield 2007). Raising gnotobiotic zebrafish until juvenile/mature stages would have required us to feed the fish using sterile live food (e.g. sterile rotifer cultures) during several weeks, without any contamination of the live cultures or the zebrafish environments, which would have been a major logistical challenge.

(3) It was not possible to keep zebrafish for more than five days post-fertilization without feeding them, as required by Laval University Animal Protection Committee.

Furthermore, the reviewer is right concerning the localization of ionocytes: At such a young age osmoregulation mostly occurs in ionocytes on the skin. In the case of zebrafish, at 96h post-fertilization, the ionocytes are densely located on the anterior part of the larvae and have started migrating towards the underdeveloped gill region (Pan *et al.* 2005). For this reason, whole larvae heads, including the gill region and the skin around it, were used for RNA extractions. This information is mentioned in the Methods section (lines 654-658):

“Before proceeding to RNA extraction, the larvae heads were isolated via dissection under microscope with sterile tools (EtOH 70%) and conserved in TRIzol™. The whole heads rather than the gills only were isolated since at that early developing stage, ionocytes of zebrafish larvae are scattered on the anterior region of the larvae and still migrating towards the underdeveloped gills (90).”

Reference

Pan, T.C., Liao, B.K., Huang, C.J., Lin, L.Y., & Hwang, P.P. Epithelial Ca(2+) channel expression and Ca(2+) uptake in developing zebrafish. *Am J Physiol Regul Integr Comp Physiol*, **289(4)**, R1202-11 (2005).

Westerfield, M. The zebrafish book, 5th Edition; A guide for the laboratory use of zebrafish (*Danio rerio*). Eugene, University of Oregon Press. Paperback. (2007).

Comment 29: Line 658-659: I understand its expensive, but you are losing a lot of variation at the individual level when pooling samples. Did you do this either as a pilot or have other publications which can verify this is an appropriate approach?

Response: The reviewer makes a valid point and we also believe it would have been interesting to get data on the interindividual variability of the metatranscriptomes. The choice to pool samples from the same sampling site for each species is a compromise to enable us to sample more sampling sites and more species. Maximizing the number of sites visited (15 sites containing 5 replicates of blackwater environments) was important to make sure that the “transcriptional response to blackwater” described in this study was not confounded with a sampling-site effect, but rather consisted of a general physiological response to blackwater. Furthermore, maximizing the number of species studied was key to document a variety of transcriptomic responses from

phylogenetically distant fish orders (Characiformes and Cichliformes) which were expected to show different physiological responses to blackwater.

Sample pooling is widespread in transcriptomic analysis: A review from Todd *et al.* 2016 on this matter showed that of 158 eco-evolutionary studies reporting statistical differential expression analysis from RNA-seq data, 89 (56%) sequenced a single library per treatment, and most single-replicate studies did use libraries constructed from pooled biological samples.

Even so, our study was not single-replicated: At least three pools of samples (one per sampling site) per species were analyzed for each water type. Analysing the transcriptomes of several sample pools, rather than a few single samples per condition, lessens the impact of single aberrant samples (Kendzioriski *et al.* 2005).

Overall, although the constraints imposed by the cost of dual RNA-Seq required us to pool individuals, our current sampling design offered the best compromise to achieve the objective of this study, i.e. “to report the responses of Amazonian fish and their associated gill microbial communities to blackwater environments”, as it enabled a general characterization of Amazonian fish responses to blackwater and prevented the occurrence of confounding effects between water type, the host species and the sampling sites.

Reference

Kendzioriski, C., Irizarry, R. A., Chen, K. S., Haag, J. D., & Gould, M. N. On the utility of pooling biological samples in microarray experiments. *PNAS*, **102(12)**, 4252–4257 (2005).

Todd, E. V., Black, M. A., & Gemell, N. J. The power and promise of RNA-seq in ecology and evolution. *Mol Ecol*, **25(6)**, 1224–1241 (2016).

REVIEWERS' COMMENTS

Reviewer #1 (Remarks to the Author):

Thank you for addressing my comments.

Reviewer #2 (Remarks to the Author):

This is a great work, and congrats to you and your co-authors on this study.